# Hematopoietic reconstitution dynamics of mobilized- and bone marrow-derived human hematopoietic stem cells after gene therapy

Mobilized peripheral blood is increasingly used instead of bone marrow as a source of autologous hematopoietic stem/progenitor cells for ex vivo gene therapy. Here, we present an unplanned exploratory analysis evaluating the hematopoietic reconstitution kinetics, engraftment and clonality in 13 pediatric Wiskott-Aldrich syndrome patients treated with autologous lentiviral-vector transduced hematopoietic stem/progenitor cells derived from mobilized peripheral blood ($n = 7$), bone marrow ($n = 5$) or the combination of the two sources ($n = 1$). 8 out of 13 gene therapy patients were enrolled in an open-label, non-randomized, phase 1/2 clinical study (NCT01515462) and the remaining 5 patients were treated under expanded access programs. Although mobilized peripheral blood- and bone marrow- hematopoietic stem/progenitor cells display similar capability of being gene-corrected, maintaining the engineered grafts up to 3 years after gene therapy, mobilized peripheral blood-gene therapy group shows faster neutrophil and platelet recovery, higher number of engrafted clones and increased gene correction in the myeloid lineage which correlate with higher amount of primitive and myeloid progenitors contained in hematopoietic stem/progenitor cells derived from mobilized peripheral blood. In vitro differentiation and transplantation studies in mice confirm that primitive hematopoietic stem/progenitor cells from both sources have comparable engraftment and multilineage differentiation potential. Altogether, our analyses reveal that the differential behavior after gene therapy of hematopoietic stem/progenitor cells derived from either bone marrow or mobilized peripheral blood is mainly due to the distinct cell composition rather than functional differences of the infused cell products, providing new frames of references for clinical interpretation of hematopoietic stem/progenitor cell transplantation outcome.

Hematopoietic Stem/Progenitor Cells (HSPC) are widely exploited in the clinic for hematopoietic stem cell transplantation (HSCT) and autologous ex vivo gene therapy (GT) for treating genetic and acquired disorders, following a preparatory conditioning regimen[1–7]. Ex vivo transduced HSPC with lentiviral vector (LV) engrafted long-term and reconstituted polyclonal hematopoiesis, achieving compelling evidence of clinical benefit and a favorable safety profile[8–14]. In particular, Wiskott-Aldrich syndrome (WAS) is a rare X-linked primary immunodeficiency associated with immune dysregulation and thrombocytopenia for which LV HSPC-

✉e-mail: aiuti.alessandro@hsr.it

GT provided a valid alternative to allogeneic HSCT, in the absence of matched donors[7,8,11].

Since a substantial fraction of the engrafting HSPC and their differentiated progeny are marked by the vector in a unique genomic site (insertion site, IS), IS analyses have been used to assess the in vivo hematopoietic dynamics after GT, unveiling common features of hematopoietic reconstitution after transplantation[15–22]. Indeed, in line with clonal tracking data in non-human primates[23–25], human hematopoietic recovery is characterized by clonal fluctuations in the early phases post-infusion of marked HSPC followed by a stable hematopoietic production[21]. In our latest IS clonal tracking study on 6 WAS subjects treated with HSPC-GT, we described that distinct HSPC subsets are responsible for the different phases of hematopoietic recovery with early myeloid production supported by myeloid progenitors present in the LV-transduced graft, while initial multi-lineage recovery is sustained by primitive Multi-Potent Progenitors (MPP). Starting from 1-2 years after GT, long-term Hematopoietic Stem Cells (HSC) become stably in charge of the hematopoietic production[18].

HSPC mobilized from the BM into the PB and harvested through leukapheresis, mobilized peripheral blood (MPB), has become in most cases the preferred method for isolating HSPC for patients undergoing HSCT[26] because of higher numbers of stem cells collected[27,28]. Collection of MPB HSPC for clinical HSCT purposes is currently based on Granulocyte-Colony Stimulating Factor (G-CSF) administration. In poor mobilizer subjects or in GT trials, G-CSF is used in combination with the C-X-C Chemokine receptor type 4 (CXCR4) antagonist Plerixafor[10,14,29–31], in order to increase the amount of retrieved HSPC. The outcome of G-CSF MPB vs. BM-based allogeneic transplantation has been characterized from the clinical point of view[32], with G-CSF MPB source displaying rapid recovery from neutropenia, reduced time of platelet-transfusion dependency and similar long-term reconstitution performance. Phenotypic profiling[33–35], cell cycle state[33,36] and gene expression[35,37,38] of BM, G-CSF MPB and Plerixafor MPB CD34+cells suggest that HSPC from distinct sources have a distinctive composition in primitive and committed progenitors as well as different functional properties, which may explain the diverse reconstitution kinetics observed in HSCT. However, the allogeneic setting is largely based on infusion of un-manipulated or partially enriched total bone marrow or mobilized leukapheresis making difficult the comparison between the biological properties of the stem cell within the different sources. In contrast, HSPC-GT offers the unique opportunity to study the behavior of purified CD34+ cells from either BM or MPB, which are infused after a short in vitro transduction protocol.

No study has evaluated so far the capability of engineered G+P MPB CD34+ cells to reconstitute the hematopoietic compartment, to survive and to maintain long-term polyclonal gene-corrected progeny, in comparison with BM-derived CD34+ cells. Given the broad application of HSPC-GT, not only in the treatment of hematological disorders[8–11,39] but also in metabolic diseases[12–14] as well as solid tumors[40–42], the evaluation of the impact of the stem cell source on the kinetic of reconstitution and on the clonality of the graft remains of importance for improving HSPC-based therapies. Thus, the main goal of this work was to dissect the behavior and fate of gene-corrected BM and mobilized HSPC subpopulations, during early hematopoietic reconstitution after GT and after reaching the steady-state condition. To this aim we performed an unplanned exploratory analysis evaluating hematopoietic dynamics in the unique model of WAS patients who received, under the same treatment scheme, CD34+ cells derived from either BM or MPB transduced with same vector and in vitro transduction protocol. Overall, these analyses provided fundamental information on the real-time contribution of different sources of HSPC to short-term and long-term engineered myelopoiesis and lymphopoiesis.

## Results

### MPB-GT patients were infused with higher amount of primitive and myeloid progenitors than BM-GT subjects

In order to assess the behavior of the two distinct HSPC sources after GT we analyzed 13 pediatric WAS-GT patients receiving either BM ($n = 5$) or MPB ($n = 7$) gene-corrected HSPC up to 3 years after GT. We also included one patient (Pt1) who received engineered HSPC from both sources. The characteristics of the patients contained in this research study are shown in Table 1 and included both patients within a clinical trial (NCT01515462) and expanded access treated subjects. BM-GT and MPB-GT cohorts of patients were treated at a comparable age and shared common features such as disease background, transduction protocol and conditioning regimen. Despite the fact that we observed a trend to higher CD34+ cell dose in the MPB-GT group, this was not statistically significant ($p = 0.06$. Fig. 1a). Moreover, we found no statistically significant difference between the two groups in vector copy number (VCN) and percentage of transduction of the infused CD34+ cells (Fig. 1b, c), suggesting similar propensity of the two sources of being gene-corrected.

However, the characterization of purified BM or MPB CD34+ cells employed for transduction[18,43] showed that BM CD34+ cells display a higher frequency of lineage-positive (LIN+) cells (Median CD34+LIN+ cells on CD34+ cells in BM vs. MPB: 65.6% vs. 13%), mainly composed by CD34+CD10+ B cell precursors, and B/NK cell progenitors (PreB/NK; Median values of PreB/NK on CD34+ cells in BM vs. MPB: 20,5% vs. 10,2%) (Fig. 1d and Supplementary Fig. 1a–e). On the other hand, MPB CD34+ cells showed higher content of myeloid subpopulations (Median values in BM vs. MPB:13,4% vs. 35,1%) (Fig. 1d) and primitive LIN-CD34+CD38- subsets (Median values in BM vs. MPB:1,2% vs. 18,5%) (Fig. 1e). Similar differences in composition were observed between BM and MPB CD34+ cells isolated from the subjects who had cells harvested from both BM and MPB (Fig. 1d, e and Supplementary Fig. 2a–d).

To address whether the culture conditions during ex vivo gene-correction could have induced distinct differentiation and/or proliferation of the two stem cell sources, we reproduced the in vitro clinical transduction protocol on BM-derived ($n = 4$) or MPB-derived ($n = 4$) CD34+ cells from WAS patients and we phenotypically characterized cultured cells. As control, we also performed in vitro transduction of CD34+ cells from BM ($n = 5$) or MPB ($n = 5$) isolated from adult healthy donors. Since the surface markers exploited for HSPC identification are not reliable after ex vivo culture due to the down-regulation of CD38 marker after culture[44,45], we exploited an alternative gating strategy that avoid the use of CD38 maker to enrich for primitive LIN-CD34^high CD45RA-CD90+ population (See Method section).

The direct comparison of BM- or MPB-derived CD34+ cell composition before transduction by using either the conventional or the alternative gating strategy, revealed that the alternative approach was not able to measure the differences in CD34+ cell composition between the two sources (Supplementary Fig. 3a–d) in both HD and WAS patients' samples. Nevertheless, BM- and MPB-derived CD34+ cells behaved similarly in response to cytokine stimulation and lentiviral exposure with the vast majority of cells maintaining their original phenotype (Supplementary Fig. 3c, d). Moreover, no statistically significant difference was observed in the proliferation rate of the two sources (Supplementary Fig. 3e).

Given these analyses, we estimated the number of infused cells for each HSPC subpopulation in BM- and MPB-GT patients based on the CD34+ cell composition assessed before transduction. We found that MPB-GT patients, were infused with higher numbers of primitive (BM vs. MPB HSC 10^6/Kg median values: 0.06 vs. 0.5; BM vs. MPB HSC+MPP 10^6/Kg median values: 0.09 vs. 1.36) and myeloid progenitors (BM vs. MPB CMP + GMP 10^6/Kg median values: 1.37 vs. 5.26) with respect to BM-GT patients (Fig. 1f–h and Supplementary Fig. 4a, b).

**Table 1 | Characteristics of the WAS-GT patients included in the study**

| | Pt1 | Pt2 | Pt3 | Pt4 | Pt7 | Pt6 | BM-Pts median values | Pt8 | Pt9 | Pt10 | Pt11 | Pt12 | Pt13 | Pt14 | MPB-Pts median values |
|---|---|---|---|---|---|---|---|---|---|---|---|---|---|---|---|
| Zhu score at baseline | 3 | 4 | 4 | 5A | 3 | 4 | 4 | 4 | 5A | 5A | 3 | 3 | 3 | 3 | 3 |
| Sex | Male | Male | Male | Male | Male | Male | NA | Male | Male | Male | Male | Male | Male | Male | NA |
| Source | BM/MPB | BM | BM | BM | BM | BM | NA | MPB | MPB | MPB | MPB | MPB | MPB | MPB | NA |
| Mobilization protocol | -/G-CSF | | | | | | NA | G-CSF | G-CSF +Plerixafor | G-CSF +Plerixafor | G-CSF +Plerixafor | G-CSF +Plerixafor | G-CSF +Plerixafor | G-CSF +Plerixafor | NA |
| Cell dose (10^6 CD34+ cells/Kg) | 3.66/5.25 | 14.1 | 10.2 | 10.3 | 7.8 | 7.8 | **10.2** | 7.0 | 16.8 | 15.5 | 11.2 | 18.4 | 26.4 | 17.6 | **16.8** |
| VCN | 1.9/1.4 | 2.4 | 2.8 | 2.3 | 4.3 | 2.3 | **2.4** | 3.2 | 3.0 | 0.9 | 3.4 | 4.0 | 2.3 | 1.5 | **3.0** |
| % of transduction (% of LV+ colonies) | 92/88 | 97 | 100 | 94 | 91 | 93 | **94** | 96 | 88 | 63 | 100 | 97 | 77 | 70 | **88** |
| Latest FU in this study (years) | 3 | 3 | 3 | 3 | 3 | 3 | **3** | 3 | 3 | 3 | 3 | 3 | 3 | 2 | **3** |

Data from Pt1 to Pt9 have been previously described[8]. Zhu clinical score was assigned to each patient before GT on the basis of clinical manifestations, according to the presence and severity of thrombocytopenia, eczema, immunodeficiency, autoimmunity and malignancies[63,66]. Pt1 is excluded from the reported Median Values of BM-Pts and MPB-Pts. Values in bold show the median value for each group of patients. (Pt patient; BM Bone Marrow; MPB Mobilized Peripheral Blood; NA Not applicable; VCN Vector Copy Number; LV lentiviral Vector; FU Follow Up).

## BM- and MPB-GT patients displayed distinct kinetics of early hematopoietic recovery but similar long-term reconstitution

To assess the hematopoietic recovery in the two groups of patients we analyzed PB and BM samples overtime after GT. In line with HSCT[29,32], MPB-GT patients showed a faster recovery of the neutrophil count (Fig. 2a). Of note, while there was no correlation between the CD34+ cell dose and the initial myeloid production, we found a positive correlation between CMP+GMP dose and the total amount of PB myeloid cells at 14 days after transplant (Fig. 2b and Supplementary Fig. 5). Although not statistically significant, these findings were also confirmed by the reduced number of days with neutrophil count <500 cell/µL in MPB-GT vs. BM-GT (Median number of days required for engraftment in BM-GT vs. MPB-GT:30 vs. 24; Fig. 2c), that negatively correlated with both total CD34+ cell and myeloid progenitor dose (Fig. 2d). We also observed a trend to a faster rise in platelet count that resulted in reduced period of being transfusion-dependent in MPB-GT patients (Fig. 2e, f). Moreover, we detected a tendency to higher platelet count in MPB-GT group starting from 1 year after GT. Similar cell counts of both myeloid and lymphoid mature compartments were observed in the two groups of patients up to 3 years after GT (Fig. 3a–e).

In line with faster hematological reconstitution, the vast majority of MPB-GT patients showed that CD34+cell count in the BM had already recovered at 1 month after GT, differently from the BM-GT group (median CD34+cells/µL: 559 in MPB-GT vs. 96 in BM-GT) (Fig. 3f, g). This tendency was also observed in HSPC subpopulations, reaching statistical significance only in the HSC population (Fig. 3h, i and Supplementary Fig. 6). Starting from 3 months after GT, both groups stabilized their CD34+ cell count reaching values similar to the pre-GT evaluation and in the range of pediatric healthy donors (HD), with comparable cell count in all BM HSPC subsets at 3 years after GT (Fig. 3f, g).

Overall, the two sources displayed different kinetics of early hematopoietic recovery, but similar long-term output. The higher number of myeloid progenitors infused in the MPB-GT group resulted in faster recovery of both myeloid and platelet counts, reducing the time of neutropenia and the number of platelet transfusions.

## MPB-GT patients showed higher transduced cell chimerism in the long term

To focus on the dynamics of gene-corrected HSPC, we performed molecular analyses on mature PB lineages, BM progenitors and HSPC subpopulations after GT.

Stable gene correction in all hematopoietic lineages was achieved in both groups starting from 12 months after GT (Fig. 4a–g and Supplementary Fig. 7a–f). We observed no statistically significant different VCN in PBMC, as well as in T and NK cells, due to the selective advantage for gene-corrected lymphocytes[46]. On the other hand, MPB-GT patients showed higher VCN with respect to BM-GT in the myeloid compartment, where no selection for transduced cells occurs, and in B cells (Supplementary Table 1). Of note, MPB-GT showed an earlier stabilization of VCN in the myeloid compartment, with constant values starting from 1 month after GT (Fig. 4d, e). Tendency of higher VCN was also detected in CD34+cell compartment (Fig. 4g) as well as in single-sorted HSPC subpopulations up to the most primitive HSC in MPB-GT patients (Fig. 4h–k and Supplementary Fig. 8a, b).

Since both groups of patients received similar sub-myeloablative conditioning[8], we reasoned that the higher VCN observed in the myeloid, B cell and CD34+cell compartments of MPB-GT group could be due to higher number of integrated copies per cell or to higher proportion of engrafted transduced cells, here defined as "transduced cell chimerism". To dissect between the two hypotheses we measured Colony Forming Cell (CFC) in BM CD34+cells purified at early (+30 days) and late (>1year) time points after treatment. By analyzing single-picked colonies, we observed similar percentage of transduced CFC

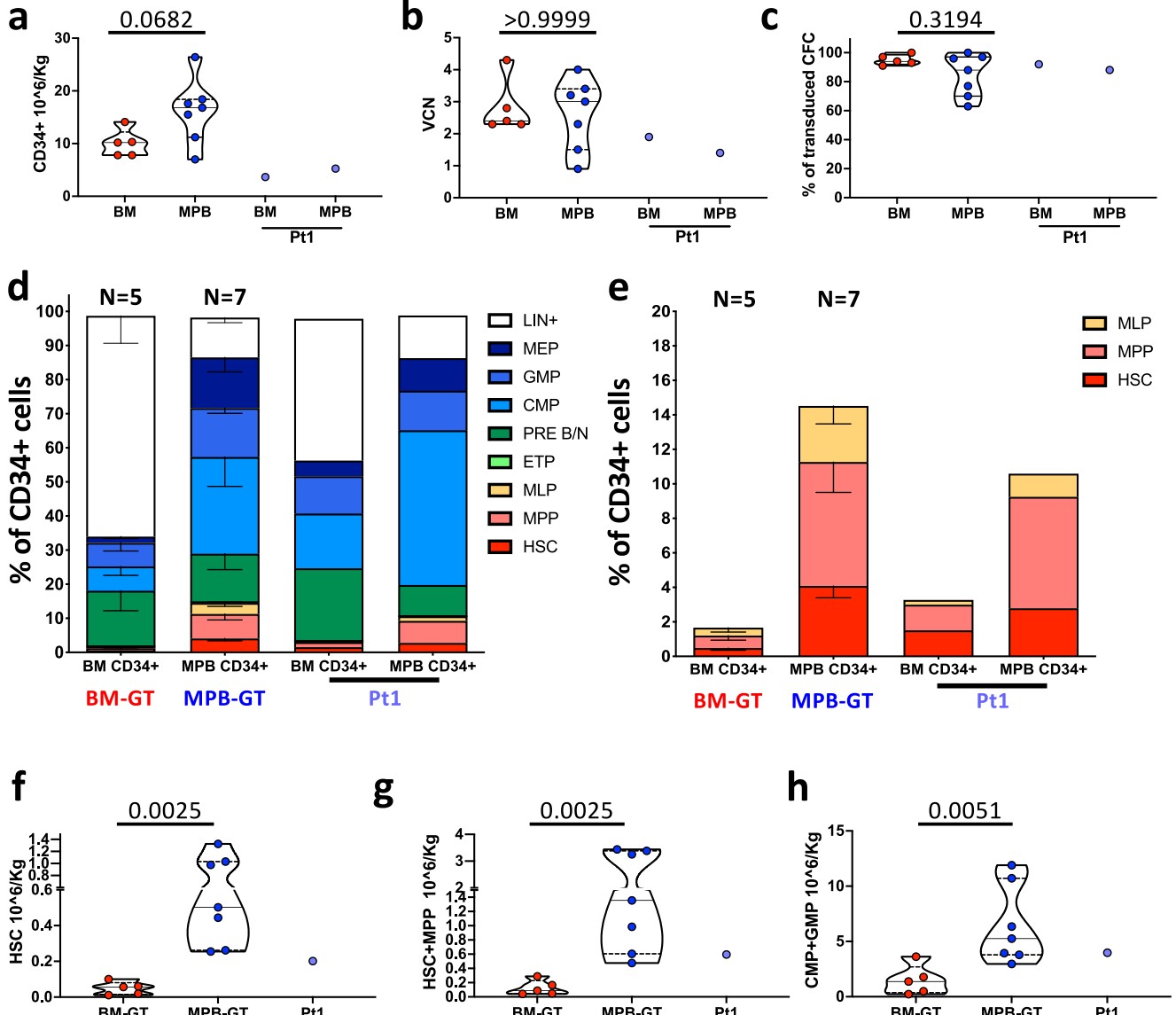

**Fig. 1 | Characterization of CD34+ cells infused in WAS BM-GT and MPB-GT patients.** Violin plots showing: **a** number of CD34+ cells/Kg infused in WAS BM-GT, WAS MPB-GT and Pt1; **b** vector copy number (VCN) estimation on a fraction of the infused CD34+ cells after 15 days of liquid culture; **c** percentage of vector positive Colony Forming Cells (CFC) derived from a fraction of the infused CD34+ cells. Lines within the violin show the median value, while dashed lines show quartile ranges. **d** Phenotypic characterization of CD34+ cells isolated from BM (for *n* = 5 BM-GT patients, biologically independent samples) and from MPB (for *n* = 7 MPB-GT patients, biologically independent samples) before transduction. Pt1 received CD34+ cells derived from both BM and MPB; both compositions are shown (see also Methods section). Data are shown as Mean +/- standard error mean **e** Focus on CD34 + LIN-CD38- compartment from BM (for *n* = 5 BM-GT patients, biologically independent samples) and from MPB (for *n* = 7 MPB-GT patients, biologically independent samples) CD34+ cells before transduction. Data are shown as Mean +/-

standard error mean. Estimation of the number of infused HSC (**f**), HSC + MPP (**g**) and CMP + GMP (**h**) subsets in BM-GT patients, MPB-GT patients and Pt1. The calculation was based on the phenotypic characterization shown in **d** and on the number of CD34+ cells infused. For Pt1 we reported the overall dose of each HSPC subpopulation from both BM and MPB sources. Lines within the violin show the median value, while dashed lines show quartile ranges (WAS Wiskott-Aldrich Syndrome; BM Bone Marrow; MPB Mobilized Peripheral Blood; GT Gene therapy; HSC Hematopoietic Stem Cells; MPP Multi-Potent Progenitors; MLP Multi-lymphoid Progenitors; ETP Early T cell Progenitors; Pre B/NK B cells and natural killer cell precursors; CMP Common Myeloid Progenitors; GMP Granulocytes-Monocytes Progenitors; MEP Megakaryocytes Erythrocytes Progenitors) (Statistical test: Two-sided Mann–Whitney; exact *p* values are shown within the graphs). Source data for panels **d** and **c** are provided as a Source Data file.

between the two groups but increased mean VCN in MPB-GT patients in the first month after GT, possibly due to the higher contribution of gene-corrected myeloid progenitors with CFC potential and high VCN (Fig. 4l, m). At late time points, MPB-GT patients displayed higher frequency of transduced colonies than BM-GT but with similar mean VCN. Of note, the frequency of transduced CFC positively correlated with the number of primitive HSC, HSC+MPP and LIN- cells infused in the patients but not with the total CD34+ cell dose (Fig. 4n–o and Supplementary Fig. 8c, d). We also found a comparable distribution in

the number of integrated copies within the single colonies at late time points, suggesting similar transduction of primitive HSPC from the two sources (Supplementary Fig. 8e–j).

Finally, since WAS patients are characterized also by platelet defects[47], we correlated the transduced cell chimerism with the platelet count at steady-state and we observed a positive correlation both at 1 year and at 2 year after GT (Fig. 4p and Supplementary Fig. 8k).

Our molecular characterization indicated that the number of primitive HSPC in the CD34+ cells before transduction plays a key role

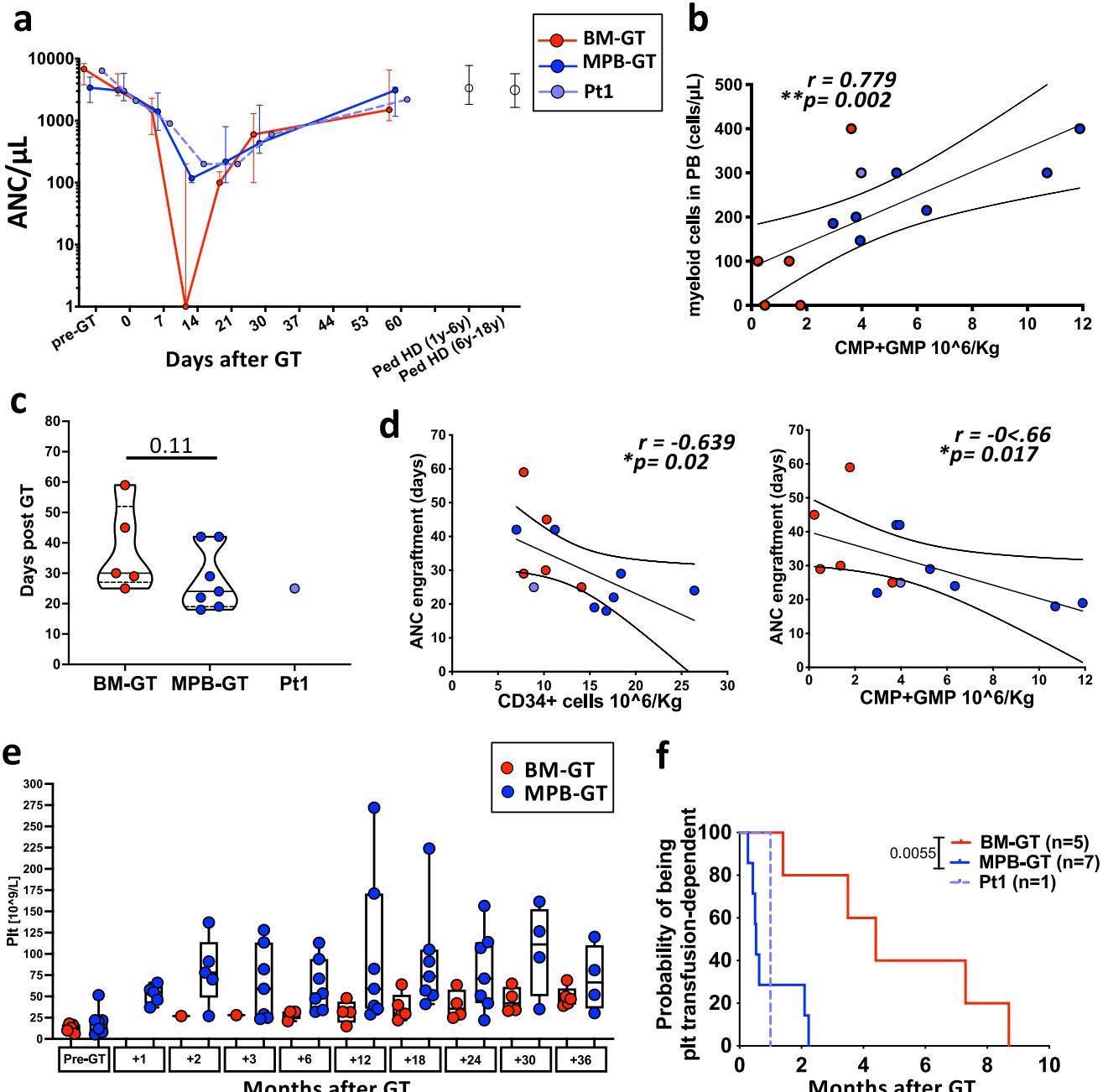

**Fig. 2 | Early myeloid and platelets reconstitution in WAS BM-GT and MPB-GT patients. a** Absolute neutrophil count (ANC) in the peripheral blood of BM-GT ($n = 5$ biologically independent samples), MPB-GT ($n = 7$ biologically independent samples) and Pt1 up to 2 months after transplantation. Data are shown as median +/- interquartile ranges. **b** Dot plots showing correlation between the number of infused CMP+GMP and the myeloid (monocytes+granulocytes) cell count 14 days after GT. Exact $p$ value is shown. Blue dots: MPB-GT patients ($n = 7$); Red dots: BM-GT patients ($n = 5$); Purple dot: Pt1. **c** Violin plot displaying day of ANC engraftment, as date of first three consecutive days with neutrophil count >500/μl, for BM-GT, MPB-GT and Pt1. Lines within the violin show the median value, while dashed lines show quartile ranges. **d** Dot plots showing correlation between (left) number of infused CD34+cells and (right) number of infused CMP+GMP with the day of ANC engraftment after GT. Exact p values are shown. **e** Box plot showing

the platelet counts at different time points after GT in BM-GT($n = 5$ biologically independent samples) and MPB-GT ($n = 7$ biologically independent samples) patients. Platelet (Plt) values are shown only in transfusion-independent patients. Boxes extend from the 25th to 75th percentiles and the whiskers mark the minimum and the maximum values. The lines in the middle of the boxes indicate the median values. **f** Graph displaying the probability of being transfusion-dependent in BM-GT and MPB-GT groups and Pt1. (BM Bone Marrow; MPB Mobilized Peripheral Blood; PB Peripheral Blood; GT Gene therapy; Ped HD Pediatric Healthy Donors; CMP Common Myeloid Progenitors; GMP Granulocytes-Monocytes Progenitors;) (Statistical test for correlations: Two-sided Spearman r. Tendency lines and 95% of confidence intervals were drawn only if residuals passed normality tests; Statistical test for curve comparison in panel f: Log-rank Matel-Cox test [**] $p$ value = 0.0055).

in the transplantation outcome. The higher number of infused HSC in MPB-GT patients was associated with higher transduced BM chimerism in vivo, ultimately leading to higher gene-correction level in HSPC, myeloid and B cell compartment and to increased platelet counts in the long-term.

**MPB-GT group showed increased number of engrafted clones**
To rule out the possibility that the higher transduced cell chimerism observed in MPB-GT patients could be the result of few engrafting clones that rapidly expanded after transplantation, we collected unique IS from PB and BM MNC samples at early (30 days),

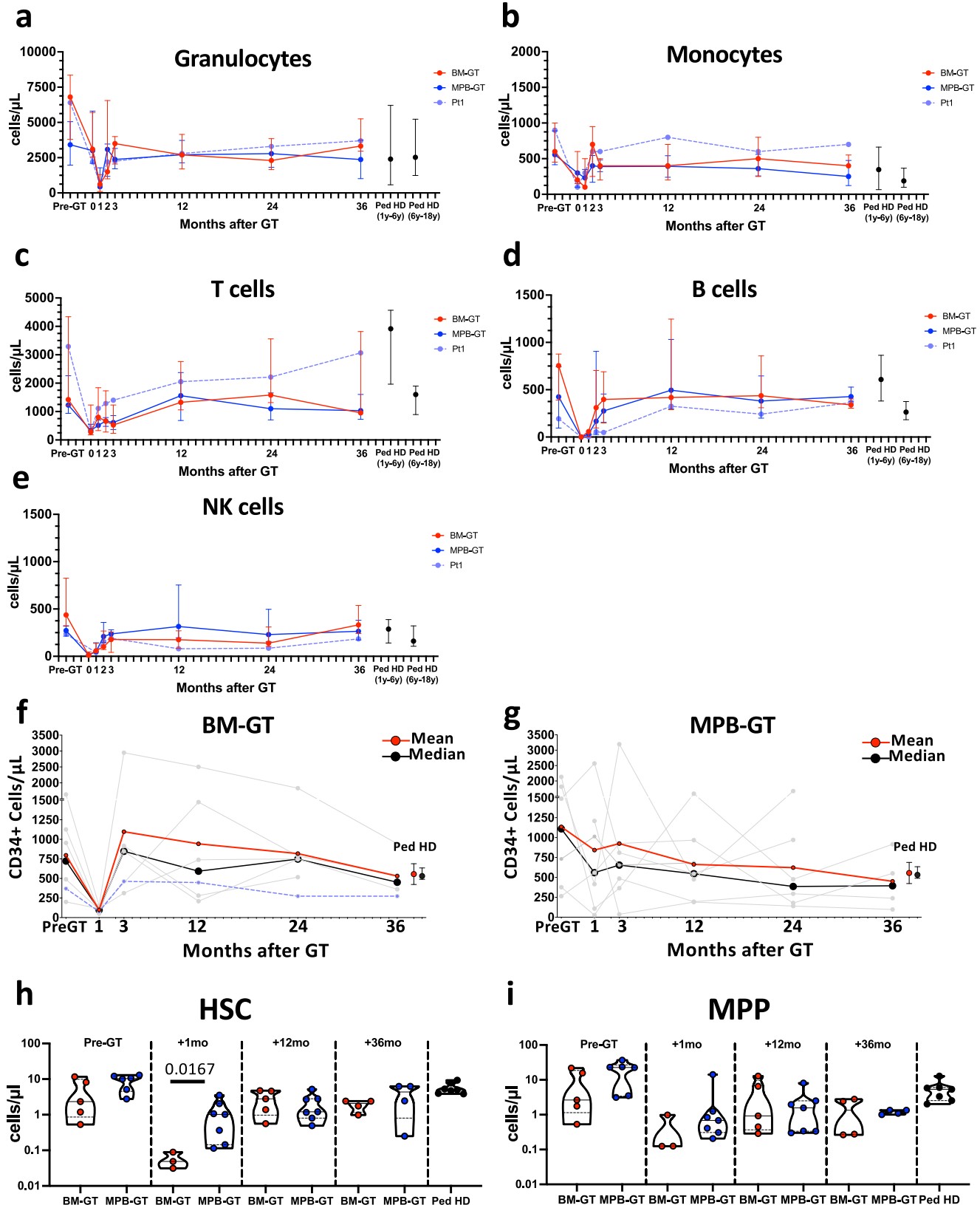

intermediate (1 year) and late (3 years) phases after GT. In addition, we collected IS from BM CD34+, PB CD3+ and PB CD15+ cells as markers of HSPC, lymphoid and myeloid compartments (Supplementary Table 2). To retrieve IS we exploited a sonication-based linker-mediated PCR method[48,49] that allows precise quantification of clonal abundance (See also Methods section).

In accordance with the higher VCN detected, we observed a tendency to higher number of IS in MPB-GT group with respect to BM-GT, that became statistically significant at steady-state condition (Fig. 5a). In addition, while both groups of patients displayed high values of Diversity Index (DI), indicating an overall polyclonality of the engrafting HSPC, the DI was higher in MPB-GT patients (Fig. 5b). These

**Fig. 3 | Long-term hematopoietic reconstitution in BM-GT and MPB-GT.** All data are shown as median +/- interquartile ranges. Peripheral blood (**a**) granulocyte, (**b**) monocyte, (**c**) T lymphocyte, (**d**) B lymphocyte and (**e**) NK cell counts in BM-GT ($n = 5$ biologically independent samples), MPB-GT ($n = 7$ biologically independent samples) and Pt1 up to 3 years after transplantation. **f, g** Graphs showing CD34+ cell count in single patients (grey lines) together with the mean (red lines) and median (black lines) values in (**f**) BM-GT group ($n = 5$ biologically independent samples) and Pt1 (purple line) and (**g**) MPB-GT patients ($n = 7$ biologically independent samples). Red and black dots display mean +/- standard deviation and median +/-

interquartile ranges values respectively of CD34+ cell counts in pediatric healthy donors (Ped HD). **h, i** Violin plots showing HSC and MPP cell counts at different time points after GT in BM-GT and MPB-GT. Black dots display values for Ped HD. Lines within the violin show the median value, while dashed lines show quartile ranges (BM Bone Marrow; MPB Mobilized Peripheral Blood; GT Gene therapy; Ped HD Pediatric Healthy Donors; NK Natural Killer; HSC Hematopoietic Stem Cells; MPP Multi-Potent progenitors) (Statistical test for groups' comparisons: Two-sided Mann–Whitney; only statistically significant $p$ values are reported within the graphs). Source Data for panels h and i are provided as a Source Data file.

data were also confirmed in the CD15+ population, while no statistically differences were observed in CD3+PB samples possibly due to the selective advantage for gene-corrected T-cells (Supplementary Fig. 9). Finally, we performed a statistical evaluation on the overall dataset to assess the variables (including cell population, time point and CD34+cell source) that mainly affect the difference in DI and IS numbers observed between the two groups of patients. We tested all possible interactions among the considered variables, and we found that MPB source has a significant additive effect on the diversity ($p = 0.0005$) and the number of IS ($p = 0.0038$) of the overall dataset, independently from the population and the timepoint analyzed (Supplementary Tables 3-4). We calculated that, using MPB as source of CD34+cells, we increased both the diversity (11.46% increase) and the number of the clones retrieved (44.36% increase) compared with BM CD34+ cells.

We then estimated the minimum number of engrafted HSPC clones by capture-and-recapture approach in myeloid cells, which are short-lived and provide a readout of stem cell output, as previously described[16,50]. Capture-and-recapture models allowed estimating the overall size of a population by accounting the number of clones observed at each independent sampling and measuring the number of shared elements (referred as re-captured clones) in subsequent timepoints. In particular, we analyzed 3 consecutive time points after GT both in early phases (<180 days) and after reaching steady-state hematopoiesis (>1 year) (Supplementary Table 5). We found a significantly higher amount of re-captured clones in MPB-GT vs. BM-GT patients, confirming the presence of increased number of gene-corrected engrafted HSPC in the BM of MPB-GT individuals (Fig. 5c, d). Of note, the number of re-captured clones at steady-state condition positively correlated with the HSC dose (Fig. 5e).

Finally, to assess the in vivo output of the engrafted primitive HSPC derived from the two sources at steady state, we measured the frequency of IS shared between sorted HSC+MPP subsets, BM progenitors and PB mature populations (Supplementary Tables 6-7). As shown in Fig. 5f, we observed similar distribution of shared IS in both groups of patients, with no skewing toward specific hematopoietic compartment. Importantly, the patient-specific level of sharing was consistent across all BM progenitors (Fig. 5g, h) and PB mature lineages (Supplementary Fig. 9d, e).

Overall, IS analyses showed that the infusion of higher number of gene-corrected primitive cells resulted in increased amount of total engrafting clones in the bone marrow of MPB-GT patients. Nevertheless, once engrafted, primitive HSPC deriving from either BM or MPB source display similar multilineage output in patients after GT at steady state.

### Primitive HSPC from BM or MPB displayed similar differentiation potential in vitro and in vivo
To further discriminate between differences in the number primitive cells vs. differences in the functional properties of the two HSPC sources, we tested the capacity of BM- or MPB-derived primitive HSPC of supporting hematopoiesis by performing in vitro and in vivo assays on sorted HSC+MPP populations from either BM ($n = 4$) or G-CSF

+Plerixafor MPB ($n = 3$) CD34+cells from healthy donors (Fig. 6a, Supplementary Fig. 10a, b).

Our in vitro differentiation assay showed that starting from the same quantity of HSC+MPP, the number of myeloid, lymphoid and erythroid cells produced were equal between the two sources (Fig. 6b). Furthermore, sorted HSC+MPP were transduced with a LV-GFP vector reproducing the ex vivo clinical protocol used for WAS HSPC gene-correction. BM or MPB-derived primitive HSPC displayed comparable frequency of GFP+ cells, similar mean VCN after 15 days of liquid culture, percentage of transduced colonies and mean VCN on single colonies (Supplementary Fig. 10c, d). Moreover, they maintained their phenotype in culture and no statistically significant difference was observed in cell proliferation after transduction (Supplementary Fig. 10e–g). We also found that MPB-derived HSC+MPP displayed higher CXCR4 expression with respect to BM-derived HSC+MPP before culture, but this difference was reduced after culture (Supplementary Fig. 10h).

After transduction, HSC+MPP from BM or MPB were transplanted at same number (10,000 cells/mouse) in NOD-SCID IL2Rγ$^{null}$ Kit$^{W41/W41}$ (NSGW41) mice (BM 10 K group $n = 9$; MPB 10 K group $n = 15$)[51,52]. Moreover, we transplanted an additional group of mice with 45,000 cells from primitive MPB (MPB 45 K group $n = 14$ mice) to reflect the distinct proportion of primitive cells measured in the two sources before cell sorting (Supplementary Fig. 10a, b).

After 12 weeks, we found that mice transplanted with equal number of BM or MPB HSC+MPP (10 K cells) showed comparable level of human CD45+ cell content and lineage differentiation capacity. We also detected a significantly higher number of human cells in the BM of MPB 45 K mice with respect to MPB 10 K group as well as in BM 10 K mice in HSPC compartment and myeloid, erythroid and mega-karyocytic lineages (Fig. 6c, d and Supplementary Fig. 10i, j). Overall, our in vivo data confirmed comparable engraftment capability and multilineage differentiation potential of primitive HSPC from both BM and MPB sources.

## Discussion
MPB is widely employed as HSPC source for ex vivo gene-correction approaches and HSCT[10,30,31,53,54]. Our analyses provide insights into the kinetic of reconstitution, longevity and lineage output of primitive and committed HSPC in 13 WAS-GT patients receiving either engineered purified MPB- or BM-HSPC. To our knowledge, this is the largest longitudinal study comparing side by side GT treatment from two HSPC sources and achieving this level of detail including deep characterization of HSPC compartment after GT, molecular analyses of hematopoietic reconstitution and IS-based clonal tracking.

Previous studies comparing total BM vs. total G-CSF MPB sources in the context of HSCT were mainly focused on the overall outcome (i.e. survival, incidence of graft versus host disease and relapse or immune reconstitution)[32,55] and commonly based on heterogeneous groups of patients with distinct type of conditioning, disease background/hematological malignancies, prior treatments and transplantation setting (i.e. autologous vs. allogeneic). No information exists on the hematopoietic reconstitution of MPB product after G-CSF+Plerixafor mobilization in comparison with BM-based HSCT. In addition, GT

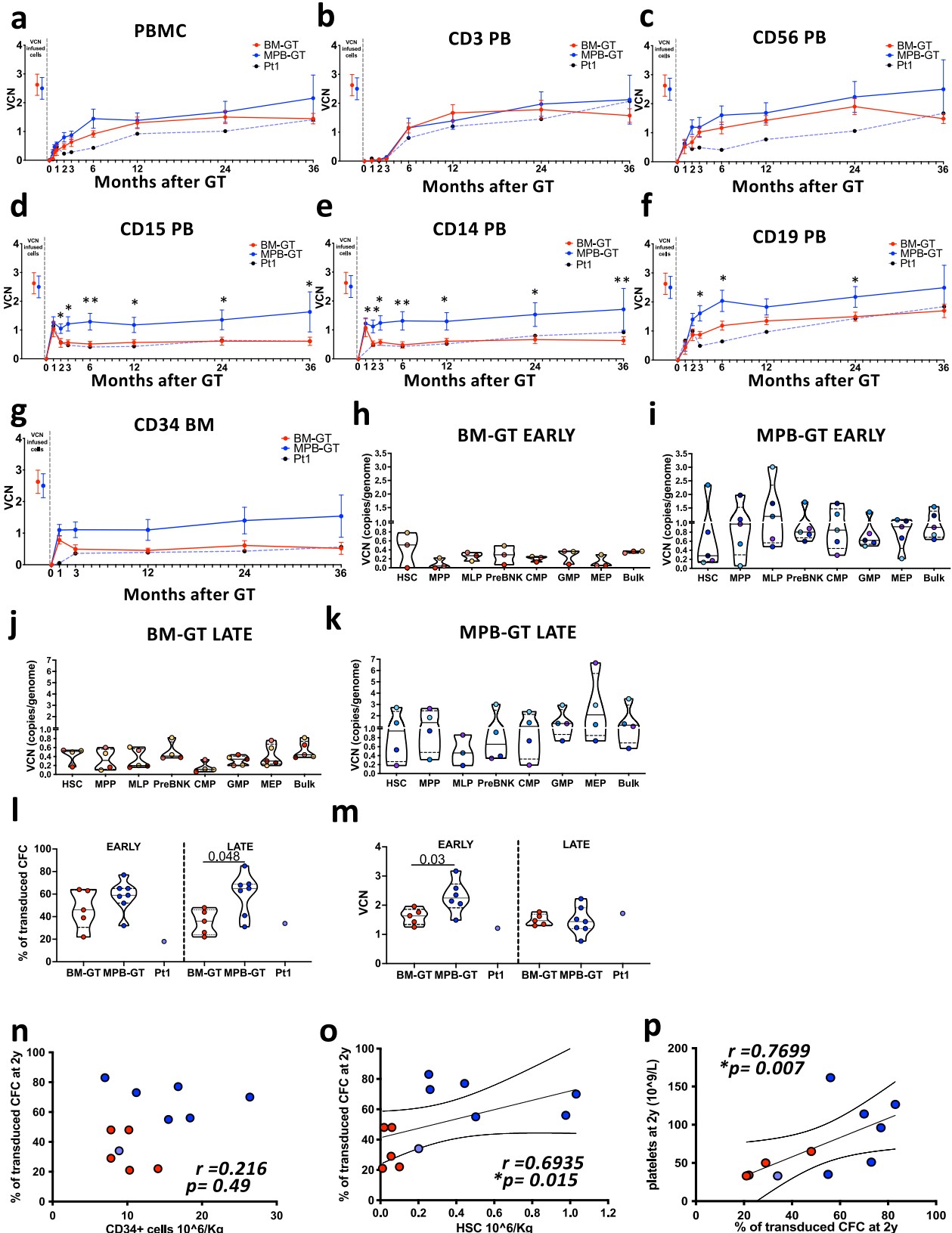

relies on purified cultured CD34+ cells while in HSCT studies HSPC are co-infused with various non-HSPC populations that influence the kinetics of engraftment. Moreover, in the clinical practice, patients undergoing HSCT are often administered post-infusion with growth factors (G-CSF) to speed up myeloid recovery[56], introducing an additional confounding factor in the kinetic of hematopoietic

reconstitution. Thus, the direct translation of the findings collected from the HSCT works to the GT field remained unexplored.

Thanks to the similarities between the two groups of WAS-GT patients in terms of disease background, transduction protocol and conditioning regimen settings we were able to assess the performance of engineered purified BM or MPB-HSPC reducing substantially the

**Fig. 4 | Gene correction of PB mature lineages, BM CD34+ cells and HSPC subsets in BM-GT and MPB-GT patients.** All data are shown as mean +/- standard error mean (SEM). Vector copy number (VCN) measurement up to 3 years after GT in (**a**) Peripheral Blood Mononuclear Cells (PBMC) and purified (**b**) CD3 PB, (**c**) CD56 PB, (**d**) CD15 PB, (**e**) CD14 PB, (**f**) CD19 PB and (**g**) CD34 BM cells in BM-GT (n = 5 biologically independent samples), MPB-GT (n = 7 biologically independent samples) and Pt1. **h–k** Violin plots displaying VCN of sorted HSPC subpopulations from BM-GT or MPB-GT patients at early (<90 days) and late (>2 years) time points after GT. Dots with the same color in all the subsets show the VCN values for the same patient. Lines within the violin show the median value, while dashed lines show quartile ranges. **l**Percentage and (**m**) mean VCN of gene-corrected colonies derived from BM CD34+ cells isolated from BM-GT patients, MPB-GT patients and Pt1 at early (30 days after GT) and late (>1 year after GT) time points. Lines within the violin show the median value, while dashed lines show quartile ranges. Dot plots showing correlation between the percentage of transduced colonies at 2 years after GT and (**n**) number of CD34+ cells infused, (**o**) number of HSC infused in GT

patients and (**p**) number of platelets at 2 years after GT. Exact p values are shown. Blue dots: MPB-GT patients (n = 6); Red dots: BM-GT patients (n = 5); Purple dot: Pt1 (BM Bone Marrow; MPB Mobilized Peripheral Blood; GT Gene Therapy; PB Peripheral Blood; HSC Hematopoietic Stem Cells; MPP Multi-Potent Progenitors; MLP Multi-lymphoid Progenitors; ETP Early T cell Progenitors; Pre B/NK = B cells and natural killer cell precursors; CMP Common Myeloid Progenitors; GMP Granulocytes-Monocytes Progenitors; MEP Megakaryocytes Erythrocytes Progenitors; CFC Colony Forming Cell) (Statistical tests on VCN collected over time: two-sided tests from longitudinal models on CD34+ cell dose, no adjustment for multiple comparisons were made $^*p < 0.05$; $^{**}p < 0.01$; exact p values were reported in Supplementary Table 1. Statistical tests for groups' comparisons: Two-sided Mann–Whitney test, only statistically significant p values are reported within the graphs. Statistical test for correlations: Two-sided Spearman r. Tendency lines and 95% of confidence intervals were drawn only if residuals passed normality tests). Source Data for panels **h**–**k** are provided as a Source Data file.

confounding factors generating several original information over the previous HSCT studies. Firstly, the faster recovery from neutropenia and reduced time for platelet transfusion dependency observed in the MPB-GT group (Fig. 2) is in line with HSCT results, and it importantly suggests that the in vitro manipulation per se does not introduce changes in the overall characteristics of CD34+ cells. In addition, one important aspect of our analyses is the correlation of the CD34+cell composition undergoing in vitro manipulation with the outcome of the gene therapy procedure. CD34 is the most used marker for determining the dose of HSPC and to enrich for HSPC in the context of HSCT and GT, despite the heterogeneity in the composition of HSPC. Our study shows that the total CD34+ cell-dose was not able to fully explain the differences in short- and long-term hematological reconstitution observed between BM- and MPB-GT patients. The two most relevant parameters were the amount of myeloid progenitors and the number of the infused primitive HSPC, positively correlating with the myeloid reconstitution (Fig. 2b) and the transduced cell BM chimerism at steady state (Fig. 4o), respectively. This suggests that a more advanced phenotypic characterization of the HSPC before ex vivo manipulation may be more predictive of the in vivo engraftment of gene-corrected cells.

In-depth phenotypic characterization was not foreseen at infusion since the markers exploited for HSPC identification are not reliable after ex vivo culture due to the down-regulation of CD38 marker[44,45], fundamental for identifying primitive HSC. Nevertheless, the preservation of the pre-transduction composition of WAS-derived CD34+ cells and HD-derived HSC+MPP (Supplementary Fig. 3 and 10) as well as the limited proliferation during the short-term culture, imply that the in vitro transduction protocol maintained the cultured population stemness without inducing differentiation.

It has been previously hypothesized that the distinct in vivo performance of the two sources were due to differences in the intrinsic properties of resident vs. mobilized HSPC, with MPB CD34+ cells displaying a more quiescent state and primitive gene signature[36,37,53]. However, these analyses did not take into consideration the diverse composition of CD34+ cells that could justify the disparity in cell cycle state and gene expression observed in the two sources. The results of our study, instead, imply that the differential behavior of the BM and MPB CD34+ cells after transplantation is mainly due to the distinct cell composition rather than functional differences of the infused cell products. Indeed, independently from the source, patients receiving similar dose of myeloid progenitors and primitive cells displayed similar kinetic of neutrophil reconstitution and BM chimerism (Figs. 2b and 4o). Moreover, our in vivo data in the humanized mouse model indicated that, when transplanted at the same number, BM- or MPB-derived primitive HSPC populations displayed comparable multilineage hematopoietic output and engraftment capability (Fig. 6 and Supplementary Fig. 10). Finally, since we cannot completely

exclude potential transcriptional differences between the two sources, we can speculate that the equal performance of primitive HSPC subsets in vivo, in both GT patients and xenotransplanted mice, might suggest that transcriptionally different populations respond similarly in a highly stressed environment, such as the one present upon transplantation.

The possibility of tracing the dynamics of infused gene-corrected HSPC through IS provided unique evidences with respect to previous HSCT reports, where there is no marking of the cells contained in the graft. Indeed, no study was conducted in order to evaluate whether the rapid hematopoietic recovery observed in MPB-based HSCT might have occurred at the expenses of the overall clonality of the hematopoietic system. In our model, we could assess that the use of MPB as source was associated with an overall higher complexity of the hematopoietic system and diversity of HSPC pool with respect to BM CD34+cells. This observation is particularly relevant in the clinical practice since it was reported that reduction of HSPC clonality lead to premature cellular senescence in HSCT[57,58,59].

Our results are in accordance with a recent work on a limited number of patients with 2 different disease backgrounds (WAS and beta-thalassemia) treated with HSPC-GT[20]. Although there were both some heterogeneity in the analyzed cohort of patients and statistically significant differences in CD34+cell dose, the authors found a higher number of active engrafted HSPC in patients treated with MPB-HSPC with respect to patients infused with gene-corrected BM-HSPC[20]. These data are confirmed by our estimation of minimum number of engrafted clones in our larger and more homogeneous cohort of WAS patients, with MPB-GT patients displaying higher number of recaptured clones at steady state (Fig. 5d).

Finally, we have observed the maintenance of a highly polyclonal pool of engineered multipotent stem cells that sustained stable production of gene-corrected cells in all hematopoietic lineages, up to 3 years after GT in all the patients analyzed (Fig. 5f–h and Supplementary Fig. 9d, e). These findings confirmed that both BM and MPB sources contain primitive and committed progenitors capable of efficiently supporting both short- and long-term hematopoiesis.

In conclusion, our results validate also from the biological point of view the preferred use of MPB as HSPC source in the context of HSPC-GT and generate new references for future interpretation of clinical data and transplantation outcome.

## Methods
### Characteristics of the WAS-GT patients involved in this research study
Our research complies with all relevant ethical regulations and it was approved by the San Raffaele Scientific Institute's Ethics Committee at IRCCS Ospedale San Raffaele. Pediatric HD samples were analyzed after parents signed informed consent approved by

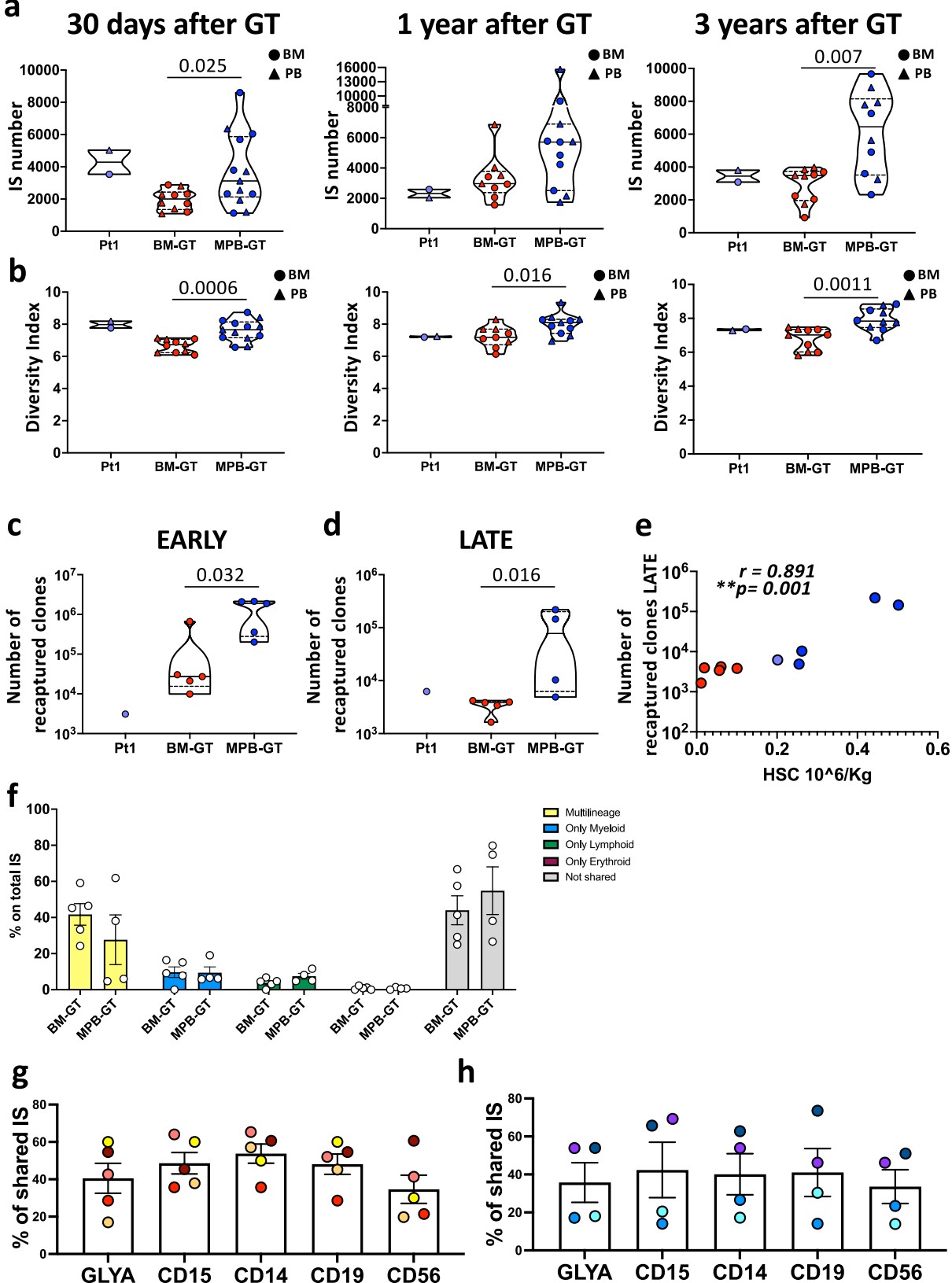

the San Raffaele Scientific Institute's Ethics Committee (TIGET09) according to CARE guidelines and in compliance with the Declaration of Helsinki principles. The sample size was determined by the number of individuals for whom exceeding BM and PB material was available in occasion of procedures performed for clinical reasons. The cohorts of pediatric HD were composed of: 7

individuals for BM evaluation (median age: 10.5 y) and 13 subjects with 1–6 years of age (median age: 4 y) and 23 subjects with 6–18 years of age (median age: 12) for PB counts. Given the rarity of BM samples from pediatric healthy individuals, sex or gender was not considered in the study design for the collection of our cohorts of PedHD for both BM samples and PB samples.

**Fig. 5 | Integration site number, Diversity index and hematopoietic output measured in WAS BM-GT and MPB-GT patients overtime.** Violin plots showing **a** the total number of Integration Sites (IS) and **b** the relative Diversity index of PB (triangles) and BM (circles) mononuclear cells collected from WAS Pt1, BM-GT and MPB-GT patients at 30 days, 1 year and 3 years after GT. IS numbers and DI have been normalized on the basis of the total amount of DNA used for retrieving IS. **c, d** Graphs showing the number of re-captured clones within myeloid populations at Early (<180 days) and Late (>1 year) time points (See also Methods section). Lines within the violin show the median value, while dashed lines show quartile ranges. **e** Dot plots showing correlation between the number of recaptured clones at Late phases after GT and number of HSC infused in GT patients. Blue dots: MPB-GT patients (*n* = 4); Red dots: BM-GT patients (*n* = 5); Purple dot: Pt1 **(f)** Frequency of IS retrieved from sorted HSC + MPP populations shared exclusively with Myeloid (CD15+ and CD14+ cells), Erythroid (Glycophorin+ cells), Lymphoid (CD3+, CD56+

and CD19+ cells) subsets or with more than one lineage belonging to different hematopoietic compartments (Multilineage) or not shared at steady state (>1 year post GT) in BM-GT (*n* = 5 biologically independent samples)or MPB-GT (*n* = 4 biologically independent samples)patients. Data are shown as Mean +/- Standard Error Mean. Frequency of IS retrieved from sorted HSC+MPP populations shared with purified mature populations at steady state (>1 year post GT) in BM-GT (*n* = 5 biologically independent samples) **(g)** or MPB-GT(*n* = 4 biologically independent samples) **(h)** patients. Data are shown as Mean +/- Standard Error Mean. Dots with the same color in all the subsets show the IS sharing levels for the same patient. (BM Bone Marrow; MPB Mobilized Peripheral Blood; PB Peripheral Blood; GT Gene Therapy; IS Integration Site; HSC Hematopoietic Stem Cells) (Statistical test for groups' comparisons: Two-sided Mann–Whitney. Exact *p* values are reported within the graphs. Statistical test for correlations: Spearman r). Source data are provided as a Source Data file.

Blood samples from WAS Pt1-9 were analyzed after parents signed informed consent approved by the San Raffaele Scientific Institute's Ethics Committee (TIGET06) at IRCCS Ospedale San Raffaele according to CARE guidelines and in compliance with the Declaration of Helsinki principles. WAS patients 1 to 9 were enrolled in an open-label, non-randomized, phase 1/2 clinical study[8], registered with Clinical-Trials.gov (number NCT01515462) and EudraCT (number 2009-017346-32).

Blood samples from WAS Pt10-14 were analyzed after parents signed informed consent approved by the San Raffaele Scientific Institute's Ethics Committee (TIGET06) at IRCCS Ospedale San Raffaele according to CARE guidelines and in compliance with the Declaration of Helsinki principles. Patients 10 to 14 were treated under early access program (compassionate use program or hospital exemption). Detailed patients' characteristics are reported in Table 1. This manuscript does not report on the primary and secondary outcome of the clinical trial which have been reported as interim analyses in the paper by Ferrua et al., Lancet Haematology 2019. The comparison between MPB and BM was not pre-specified in the study protocol and it has been performed as additional exploratory analyses for the purpose of this publication and approved by the San Raffaele Scientific Institute's Ethics Committee at IRCCS Ospedale San Raffaele.

In order to analyze the groups of BM-GT vs. MPB-GT patients with the same range of age, we included in this study all the pediatric GT patients treated since 2010 up to 2018, with the exception of one patient who shows an undetectable CD90 marker expression, likely due to modification in the protein conformation (currently under investigation). In particular, the ranges of age at treatment were from 1.9 and 5.9 years (median age:1.9) for the BM-GT group and from 1.4 to 14.4 years (median age: 10.3) for MPB-GT patients. The data cutoff was chosen to evaluate all the available patients up to 2 year-follow up. The choice of the source was based on the evaluation of different clinical factors including age and weight of the patient, estimated yield of CD34+ cells (according to the content of bone marrow CD34+ cells evaluated by previous bone marrow aspiration, performed at screening) and feasibility of the procedure. Moreover, a mixture of MPB and BM CD34+cells was foreseen if required to obtain a sufficient cells dose. In particular, Pt1 required administration of cell drug product from both cell sources for reaching optimal cell dose. For Pt#2-7 (BM-GT) and for Pt#8-9 (MPB-GT) the source was chosen due to the age of the patients. Indeed, BM harvest was the preferred choice for younger pediatric subjects at the time of the procedure. With the introduction of the Plerixafor in the mobilization protocol, the improvement of the clinical management of the leukapheresis procedure as well as the CD34+cell yield obtained in patients Pt#8-9, the remaining patients treated in the expanded access program (Pt#10-14) were treated with MPB CD34+cells. All the described patients are alive, clinically well and no adverse event related to the product has been observed to date. All available

healthy donors' and patients' samples are reported in the manuscript. For clinical data, source data include all the documents related to the patient during the trial or patient's treatment under expanded access (infused CD34+ cell dose, VCN and percentage of transduction of the infused cell dose, blood cell counts, requirement for platelet transfusion, VCN in PB and BM subpopulation, VCN and percentage of transduction on BM CFC); these data were collected at IRCCS Ospedale San Raffaele (Milan, Italy) and entered in the respective case report form and monitored according to the clinical trial/expanded access protocol and local guidelines. All the other exploratory research data (flow cytometry and integration site analyses) were collected at San Raffaele Telethon Institute for Gene Therapy and they were analyzed and stored in a dedicated database. 6 out of 14 patients (Pt1 to Pt7) included in this research study were already present in our previous published work[18], however novel datasets were generated for this manuscript, as mentioned in the specific Methods sections.

## In vivo models

Mouse studies were conducted according to protocols approved by the San Raffaele Scientific Institute and the Italian Ministry of Health (IACUC, #1091). NOD.Cg-Kit[W-41J] Prkdc[scid] Il2rgtm1Wjl/[WaskJ] (NSGW41, stock #026497) mice were purchased from the Jackson Laboratory. All animals were maintained in the SPF animal facility at IRCCS Ospedale San Raffaele with no more than 5 animals per cage (cage covered with filter, sterile air ventilation) at a temperature in the range 20 to 24 °C and a relative humidity of 45–65%; at approximately one cycle of light (12 h light and 12 h dark) with ad libitum food and water. The type of diet used was VRF1(P), a GLP certificated rat and mouse breeding diet, containing elevated levels of heat-labile vitamins, which make it suitable for autoclaving and for animals with high vitamin requirements (i.e. SPF/Germ free). The handling of the animals was performed by trained personnel, with the aim of minimizing the degree of stress and suffering and the period of constraint. The degree of well-being and the clinical conditions of the animals were assessed by daily post-treatment or transplantation observation. During detailed clinical observation the following parameters were evaluated: loss of body weight, posture, activity level, fur and skin appearance, breathing and reaction to manipulation. All animals were euthanized by $CO_2$ inhalation as described in our specific authorized research project and carried out following the AVMA Guidelines for the Euthanasia of Animals: 2020 Edition.

## Flow cytometry analyses on patients' samples

Identification and quantification of HSPC subpopulations from BM-GT and Pt1 were previously performed[18]. MPB-GT patients' PB and BM samples as well as the BM of transplanted mice were analyzed using Whole Blood Dissection (WBD) cytometry assay[43]. In brief, after red-blood cell lysis, the samples were labeled with the following fluorescent antibodies:

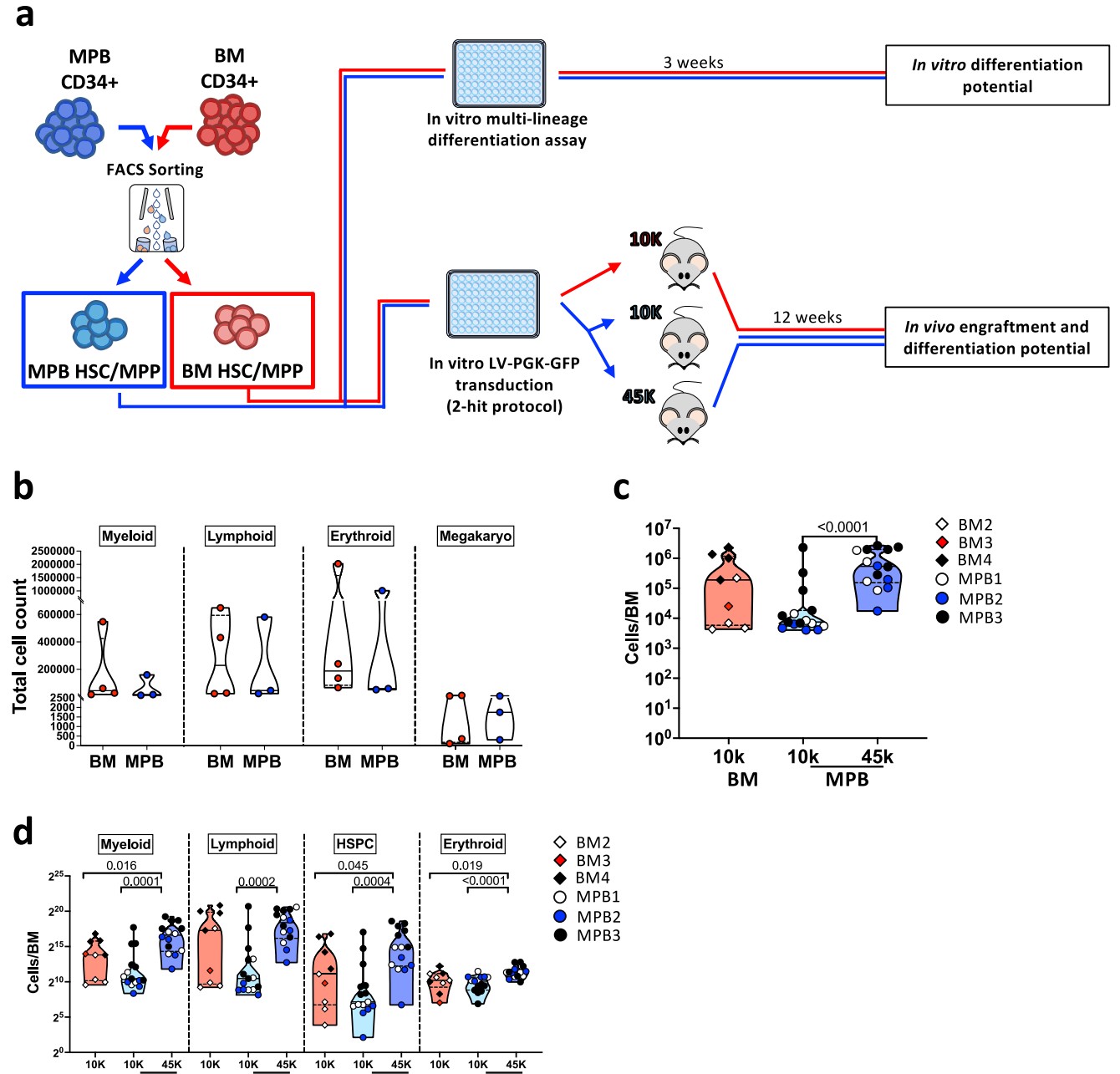

**Fig. 6 | In vitro and in vivo characterization of HSC+MPP population from BM and MPB CD34+ cells. a** Experimental scheme for in vitro and in vivo characterization of purified LIN-CD34+CD38-CD45RA- (HSC+MPP) populations. After sorting, 500 HSC+MPP cells/well from BM ($n = 4$ donors) and MPB ($n = 3$ donors) sources were cultured for 3 weeks in a medium containing a cocktail of cytokines to assess differentiation in all major hematopoietic lineages (myeloid, lymphoid, erythroid and megakaryocyte) (See also Methods section). The remaining sorted populations were pre-stimulated for 16 h and transduced with a PGK-GFP lentiviral vector, following the 2-hit protocol used in the WAS-GT clinical trial. After transduction, 10,000 cells from BM (BM 10 K, $n = 3$ donors), 10,000 cells from MPB (MPB 10 K, $n = 3$ donors) and 45,000 cells from MPB (45 K MPB, $n = 3$ donors) were transplanted in 6-7 week-old NSGW41 mice. After 12 weeks, mice were euthanized and human BM content was analyzed to assess engraftment and in vivo differentiation of primitive cells from BM or MPB source. **b** Number of myeloid, lymphoid, erythroid and megakaryocyte differentiated cells from 500 BM ($n = 4$ donors) or MPB ($n = 3$ donors) HSC+MPP populations after 3 weeks of culture. Each dot shows a distinct donor. **c** Absolute count of total human CD45+ cell content in the BM of 10 K BM, 10 K MPB and 45 K MPB mice 12 weeks after transplantation. **d** Absolute count of human myeloid, lymphoid, HSPC and erythroid cell content in the BM of 10 K BM, 10 K MPB and 45 K MPB mice 12 weeks after transplantation. Lines within the violin show the median value, while dashed lines show quartile ranges (BM Bone Marrow; MPB Mobilized Peripheral Blood; HSC Hematopoietic Stem Cells; MPP Multi-Potent Progenitors; FACS Fluorescent Activated Cell Sorter; LV Lentiviral Vector; PGK phosphoglycerate kinase; GFP Green Fluorescent Protein) (Statistical test for groups' comparisons: Two-sided Mann–Whitney. Only statistically significant p values are reported within the graphs). Source data are provided as a Source Data file.

- Mouse anti-human CD3-BV605 (Clone: OKT3; Biolegend, 317322), Verified Reactivity: Human; Application: Flow cytometric analysis of antibody surface-stained cells. Dilution: 1:50

- Mouse anti-human CD56-PC5 (Clone: 5.1H11; Biolegend, 362516), Verified Reactivity: Human; Application: Flow cytometric analysis of antibody surface-stained cells. Dilution: 1:50

- Mouse anti-human CD14-BV510 (Clone: M5E2; Biolegend, 301842), Verified Reactivity: Human, Cynomolgus, Rhesus; Application: Flow cytometric analysis of antibody surface-stained cells. Dilution: 1:50
- Mouse anti-human CD33-BB515 (Clone: WM53; BD Biosciences, 564588), Verified Reactivity: Human (QC Testing); Application: Flow cytometry (Routinely Tested). Dilution: 1:50
- Mouse anti-human CD41/CD61-PC7 (Clone: A2A9/6; Biolegend, 359812), Verified Reactivity: Human; Application: Flow cytometric analysis of antibody surface-stained cells. Dilution: 1:50
- Mouse anti-human CD66b-BB515 (Clone: G10F5; BD Biosciences, 564679), Verified Reactivity: Human (QC Testing); Application: Flow cytometry (Routinely Tested). Dilution: 1:50
- Mouse anti-human CD7-BB700 (Clone: M-T701; BD Biosciences, 566488), Verified Reactivity: Human (QC Testing), Rhesus, Cynomolgus, Baboon (Reported); Application: Flow cytometry (Routinely Tested). Dilution: 1:50
- Mouse anti-human CD45-BUV395 (Clone: HI30; BD Biosciences, 563792), Verified Reactivity: Human (QC Testing); Application: Flow cytometry (Routinely Tested). Dilution: 1:33
- Mouse anti-human CD38-BUV737 (Clone: HB7; BD Biosciences, 612824), Verified Reactivity: Human (QC Testing); Application: Flow cytometry (Routinely Tested). Dilution: 1:33
- Mouse anti-human CD90-APC (Clone: 5E10; BD Biosciences, 559869), Verified Reactivity: Human (QC Testing), Rhesus, Cynomolgus, Baboon, Pig, Dog (Tested in Development); Application: Flow cytometry (Routinely Tested). Dilution: 1:33
- Mouse anti-human CD135-PE (Clone: BV10A4H2; Biolegend, 313306), Verified Reactivity: Human; Application: Flow cytometric analysis of antibody surface-stained cells. Dilution: 1:33
- Mouse anti-human CD11c-BV650 (Clone: B-ly6; BD Biosciences, 563404), Verified Reactivity: Human (QC Testing); Application: Flow cytometry (Routinely Tested). Dilution: 1:20
- Mouse anti-human CD10-BV786 (Clone: HI10a; BD Biosciences, 564960), Verified Reactivity: Human (QC Testing), Rhesus, Cynomolgus, Baboon (Tested in Development); Application: Flow cytometry (Routinely Tested). Dilution: 1:20
- Mouse anti-human CD34-BV421 (Clone: 561; Biolegend, 343610), Verified Reactivity: Human; Application: Flow cytometric analysis of antibody surface-stained cells. Dilution: 1:20
- Mouse anti-human CD45RA-APCH7 (Clone: HI100; Biolegend, 304128), Verified Reactivity: Human; Reported Reactivity: Chimpanzee; Application: Flow cytometric analysis of antibody surface-stained cells. Dilution: 1:20
- Mouse anti-human CD71-BV711 (Clone: M-A712; BD Biosciences, 563767), Verified Reactivity: Human (QC Testing); Application: Flow cytometry (Routinely Tested). Dilution: 1:20
- Mouse anti-human CD19-APCR700 (Clone: SJ25C1; BD Biosciences, 659121), Verified Reactivity: Human; Application: Flow cytometry. Dilution: 1:20

All the antibodies were purchased from Biolegend and BD Biosciences and they are well-characterized and validated by providers. Titration assays were performed to assess the best antibody concentration. After surface marking, the cells were incubated with PI (Biolegend) to stain dead cells. Absolute cell quantification was performed by adding Flowcount beads (BD Biosciences) to samples before WBD procedure. All stained samples were acquired through BD LSR-Fortessa or BD Symphony A5 (BD Biosciences) cytofluorimeters after Rainbow beads (Spherotech) calibration. Raw data were collected through DIVA software Version 8.0.2 and analyzed with FlowJo software Version 10.5.3 (BD Biosciences). Technically validated results were always included in the analyses, and we did not apply any exclusion criteria for outliers.

**In vitro transduction of HD and WAS patients' derived BM or MPB CD34+ cells**
BM CD34+ cells isolated from HD were purchased from Lonza (2M-101C), while MPB G+P CD34+ cells isolated from healthy subjects were

purchased from StemExpress (MLEGP34005C). Frozen BM CD34+ cells before GT from Pt2, Pt3, Pt4 and Pt9 and frozen MPB CD34+ cells before GT from Pt8, Pt9, Pt12 and Pt14 were thawed to perform in vitro transduction experiments. An aliquot of cells from both HD and MPB, before transduction, was labeled with:

- Mouse anti-human Lineage cocktail (anti-CD3/CD14/CD16/CD19/CD20/CD56)-BV510 (Clones: OKT3, M5E2, 3G8, HIB19, 2H7, HCD56; Biolegend, 348807), Verified Reactivity: Human; Application: Flow cytometric analysis of antibody surface-stained cells. Dilution: 1:10
- Mouse anti-human CD15-BV510 (Clone: W6D3; Biolegend, 323028), Verified Reactivity: Human; Application: Flow cytometric analysis of antibody surface-stained cells. Dilution: 1:50
- Mouse anti-human CD34-PB (Clone: 581; Biolegend, 343512), Verified Reactivity: Human; Reported Reactivity: Cynomolgus; Application: Flow cytometric analysis of antibody surface-stained cells. Dilution: 1:20
- Mouse anti-human CD38-PC5 (Clone: HIT2; Biolegend, 303508), Verified Reactivity: Human; Reported Reactivity: Chimpanzee, Horse, Cow; Application: Flow cytometric analysis of antibody surface-stained cells. Dilution: 1:50
- Mouse anti-human CD10-PC7 (Clone: HI10a; Biolegend, 312214), Verified Reactivity: Human, Cynomolgus, Rhesus; Reported Reactivity: African Green, Baboon, Capuchin monkey, Chimpanzee; Application: Flow cytometric analysis of antibody surface-stained cells. Dilution: 1:33
- Mouse anti-human CD7-APCR700 (Clone: M-T701; BD Biosciences, 659124), Verified Reactivity: Human; Application: Flow cytometry. Dilution: 1:20
- Mouse anti-human CD135-PE (Clone: BV10A4H2; Biolegend, 313306), Verified Reactivity: Human; Application: Flow cytometric analysis of antibody surface-stained cells. Dilution: 1:33
- Mouse anti-human CD45RA-APCH7 (Clone: HI100; Biolegend, 304128), Verified Reactivity: Human; Reported Reactivity: Chimpanzee; Application: Flow cytometric analysis of antibody surface-stained cells. Dilution: 1:20

All the antibodies were purchased from Biolegend and BD Biosciences and they are well characterized and validated by providers.
Here below the alternative gating strategies.

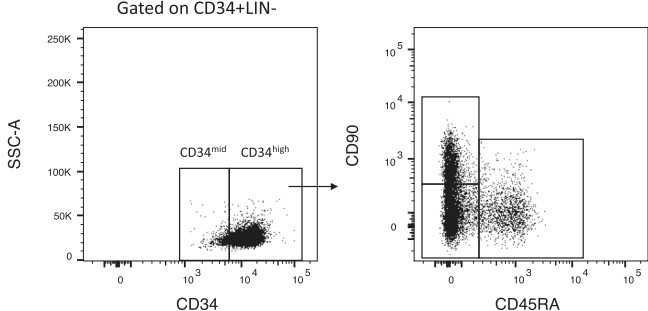

Gated on CD34+LIN-

Briefly, after gating on CD34+LIN- cells, we identify CD34$^{mid}$ and CD34$^{high}$ populations. Within CD34$^{high}$ cells, we identified on CD45RA+, CD45RA-CD90- and CD45RA-CD90+ populations. Remaining cells were resuspended at a cell density of $1 \times 10^6$ cells/ml in serum-free CellGro medium (Cell Genix) containing human SCF (300 ng/ml), TPO (100 ng/ml), FLT3 (300 ng/ml), IL-3 (60 ng/ml) all from Peprotech. Cells were then plated in RetroNectin (Takara)-precoated wells. After 16 h of pre-stimulation, cells were transduced with lab-grade GFP-LV (2 rounds of transduction, each lasting 24 h). After transduction, cells were stained with anti-human LIN cocktail (anti-CD3/CD14/CD16/CD19/CD20/CD56), anti-CD15, anti-CD34, anti-CD45RA (Biolegend) and anti-CD90 (BD Biosciences), antibody specification described above, as well as with Mouse anti-human CD184 (CXCR4)-PE (Clone: 12G5; Biolegend, 306506; Verified Reactivity: Human, Cynomolgus,

Rhesus; Application: Flow cytometric analysis of antibody surface-stained cells. Dilution:1:50).

All stained samples were acquired through BD Symphony A5 (BD Biosciences) cytofluorimeters after Rainbow beads (Spherotech) calibration Raw data were collected through DIVA software Version 8.0.2 and analyzed with FlowJo software Version 10.5.3 (BD Biosciences). The graphical output was generated through Prism v9.1.0 (GraphPad software). Technically validated results were always included in the analyses, and we did not apply any exclusion criteria for outliers.

### Isolation of PB mature lineages, BM progenitors and HSPC subpopulations

PB and BM mononuclear cells from WAS-GT patients were isolated using Ficoll-Hypaque gradient separation (Lymphoprep, Fresenius). Mature PB lineages and BM progenitors were purified using positive selection with immunomagnetic beads (average purity, 94.6%) according to the manufacturer's specifications (Miltenyi Biotec). HSPC subpopulations were FACS purified with the MoFlo XDP cell sorter (Beckman Coulter) as previously described[18], achieving purity ranging between 92% and 99%.

### Determination of VCN by droplet digital PCR

Genomic DNA was extracted with the QIAamp DNA Blood Mini or Micro Kit (QIAGEN), and whole-genome amplification was performed with the Repli-g Mini Kit (QIAGEN) on DNA from FACS-sorted HSPC subpopulations, according to the manufacturer's specifications To evaluate the number of lentiviral vector copies integrated per genome, the droplet digital (ddPCR) technique was used as previously described[18]. In brief, ddPCR assay is based on a primers/probe set designed to detect DNA sequences on the common packaging signal region of LV (HIV system). To normalize for the exact amount of template used in each reaction, an endogenous control assay is set up using a DNA sequence specific to a region of the human *GAPDH* gene (GAPDH system). The target and reference molecule concentration is calculated in an end-point measurement that enables the quantification of nucleic acids without the use of standard curves and independent of reaction efficiency. The VCN is determined by calculating the ratio of the target molecule concentration to the reference molecule concentration, times the number of copies of reference species in the genome. All the reactions were performed according to the manufacturer's instructions and analyzed with a QX200 Droplet Digital PCR System (software: QuantaSoft Version1.7.4.0917) (Biorad).

### Integration site analyses

**Integration sites retrieval.** IS from PB mononuclear cells, BM mononuclear, BM CD34+, PB CD3+ and PB CD15+ cells isolated from all the patients were collected through SLIM-PCR, as previously described[60], that combines fragmentation of genomic DNA by sonication and tagging of the IS-containing genomic fragments with random barcodes prior PCR amplification. Briefly, extracted DNA undergoes mechanical fragmentation through sonication followed by a reaction of ligation with a barcoded linker cassette to label each sample with specific identifier. Barcoded linker cassettes also include a small region with random nucleotides that allow tagging differently each single fragment of a sample. IS dataset from HSPC and mature lineages isolated from BM-GT patients and Pt1 for evaluating the hematopoietic output of primitive HSPC subsets in vivo (Fig. 5f, g and Supplementary Fig. 9d) was previously described[18]. For the generation of MPB-GT patients' dataset IS were collected through LAM-PCR (for Pt8 and Pt9) and SLIM−PCR (for Pt10, Pt11) coupled with high-throughput Illumina sequencing[48,49].

**Integration sites mapping and filtering.** All the datasets, including the BM-GT IS previous published in Scala et al. 2018, were re-mapped with the most recent software VISPA2[49] to homogeneously analyze IS

between the two group of patients. To minimize any potential cross-contamination, we extended our previous approach, described in[16,46], introducing the attribution of each single IS to a patient based on the date of sample processing: whenever an IS is shared between two patients, we firstly consider whether the same IS was previously assigned to one of the two subjects, then we applied the 10 fold rule presented in[16,46] that assigns the IS to the patient with higher sequence count. Finally, we removed all the IS with a sum of sequencing count below 3 in the entire dataset.

**Normalization of IS numbers and Diversity Index on DNA amount.** Potential limitations of entropy-based measures are related to the heterogeneous nature of next-generation sequencing data[61]. Shannon entropy index (h-index) rapidly increases with the amount of the DNA until it reaches a steady.

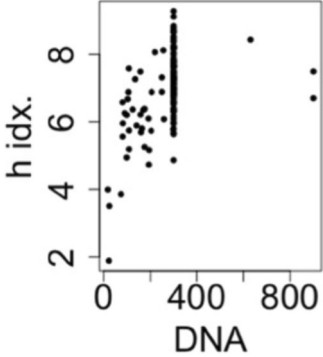

Shannon enotropy index (y-axis) as a function of the host DNA amount for the WAS clinical trial under study

To remove this effect, we exploited a Shape-constrained splines (SCS) normalization approach[62]. In particular, we firstly described the logarithmic Shannon entropy index (or the logarithmic n. of IS) as a function of the DNA amount, and the residuals were used as the rescaled values.

**Exploratory analysis for variable interactions.** To best identify how the variables interact each other (for example "*mrk*" identifying the cell marker, "*source*" as HSPC source, and "*t*" as follow-up timepoint), we first analyzed the behavior of the logarithmic dataset entropy by comparing a set of linear models each featuring different combinations of variable interactions, such as "*mrk*"X"*source*", "*mrk*"X"*t*", "*source*"X"*t*" and "*mrk*"X"*source*"X"*t*". Moreover, we always included a linear effect of the DNA amount processed for IS retrieval and the VCN to consider their effect on the clonality level. According to the Akaike Information Criterion (AIC), as benchmarking score for best candidate model selection, the resulting best model for our dataset selected the interaction "*mrk*"X"*t*", plus an additive effect of the source ($log(h)$ = ($l$) + "*DNA*" + "*VCN*" + "*mrk*" + "*t*" + "*source*" + "*mrk*"X "*t*"), with AIC = −96,58792 (Supplementary Table 3). Moreover, we found that the effect of the stem cell source was statistically significant (*t*-test; $p = 0.0005$).

Thus, we estimated that MPB has overall a positive impact on the clonal diversity with a rate of change (roc) equal to

$$roc = \frac{e^{\beta_{mrk \times t} mrk \times t + \beta_{MPB} + \beta_{DNA}DNA + \beta_{VCN}VCN}}{e^{\beta_{mrk \times t} mrk \times t + \beta_{DNA}DNA + \beta_{VCN}VCN}} = e^{\beta_{MPB}} = 1.1146$$

Corresponding to an increase in Shannon entropy of 11.46% than samples collected from BM CD34+. We performed the same analytical approach using the number of distinct clones in log scale as response variable. In this context, we found that the best model (presenting the lowest AIC, AIC = 262, 8259, Supplementary Table 4) selected the same variables and interactions ($log(nIS)$ = ($l$) + "*DNA*" + "*VCN*" + "*mrk*" + "*t* "+

" *source*" + "*mrk*"*X*"*t*"). Similarly to what previously identified, the effect of the source is statistically significant (*t*-test; *p* = 0.0038) and the number of clones in MPB samples shows an increase of 44.36% than samples derived from BM. The mathematical model identified a positive impact in the number of distinct clones with a rate of change equal to e^(β_MPB)=1.4436, corresponding to 44.36% in favor to MPB than BM source.

**Estimation of population abundance by capture-recapture approach.** The estimation of population abundance was performed by capture re-capture statistics as reported[16,46]. The capture re-capture models estimate the overall size of a population by accounting the number of elements observed at each independent sampling and measuring the amount of shared elements among captures. We applied the well-established Chao1 model[50] using IS retrieved in Myeloid cells (composed by IS retrieved from PB CD14+ and PB CD15+ samples) and we used 3 consecutive timepoints after GT: 30, 60 and 90 days for EARLY phase and 12, 24 and 36 months for LATE phase (Supplementary Table 5). From our assumptions, we used closed population requirements, although we also tested for open population requirements without major difference. Abundance estimations Chao1 and their standard errors were calculated using the R package Rcapture[64] (Version 1.4-3) with the function *M* for log-linear models and closed populations *closedp*.

**Isolation and in vitro transduction of sorted primitive HSPC from BM or MPB sources**
Cells were labeled with anti-human LIN cocktail (anti-CD3/CD14/ CD16/CD19/CD20/CD56), anti-CD15, anti-CD34, anti-CD38, anti-CD45RA (Biolegend) and anti-CD90 (BD Biosciences), antibodies specification described above, to perform purification of Primitive subpopulations (Lin- CD34+CD38- CD45RA- cells). The stained samples were FACS purified with the BD Aria cell sorter (BD Biosciences), achieving purity ranging between 92% and 99%.

Sorted primitive (HSC+MPP) populations were transduced as described above in "In vitro transduction of HD and WAS patients' derived BM or MPB CD34+ cells" section. After transduction, cells were collected, washed, and transplanted in NSGW41 mice. An aliquot of cells was cultured in Iscove's modified Dulbecco's medium (IMDM), 10% fetal bovine serum (Cambrex, East Rutherford, NJ, USA) with stem cell factor (SCF), FLT3-L thrombopoietin (TPO) and IL-3 (all from Preprotech) at 20 ng/ml concentration (liquid culture) and harvested after 15 days to perform VCN estimation. An additional aliquot was used to perform Colony Forming Cell (CFC) assay according to the manufacturer's procedure in Methocult medium (Stem Cell Technologies, Vancouver, Canada). At day 14, colonies were scored, singly picked and analyzed to evaluate the percentage of transduction.

**In vivo transplantation assay**
Transduced primitive (HSC+MPP) subsets from either BM or MPB source were adoptively transferred into eight-week-old female NSGW41 mice (BM 10 K and MPB 10 K groups=1 × 10^4/200 μl/mouse, MPB 45 K group=4.5 × 10^4/200 μl/mouse). Human cell engraftment in murine BM was assessed at 12 weeks after transplantation, by applying WBD protocol (see "Flow cytometry analyses on patients' samples" section).

**In vitro multi-lineage differentiation assay**
In vitro assay was performed in non-tissue culture-treated 96-well flat bottom plate (Falcon). 2 h before cell seeding, plates were coated with StemSpan Differentiation Coating Material (Stem Cell Technologies) according to manufacturer specifications. Primitive subpopulations (Lin- CD34+CD38- CD90+CD45RA- and Lin- CD34+CD38- CD90- CD45RA-) derived from either BM or MPB sources, were seeded in SFEM II medium (Stem Cell Technologies)

supplemented with hSCF (100 ng/ml), hFLT3 (10 ng/ml), hIL-7 (100 ng/ml), hIL-2 (10 ng/ml) (Novartis), hTPO (75 ng/ml), hIL-6 (40 ng/ml), hIL-3 (10 ng/ml), hIL-11 (50 ng/ml), hEPO (0.1 U/ml) (Peprotech), hIL-4 (10 ng/ml) (Miltenyi Biotec), hLDL (4 μg/ml) (Stem Cell technologies). Medium change was performed every 3-4 days. After 3 weeks of culture, cells were harvested and labeled with the following anti-human antibodies:

- Mouse anti-human CD235a-PE (Clone: GA-R2/HIR2; BD Biosciences, 561051), Verified Reactivity: Human (QC Testing); Application: Flow cytometry (Routinely Tested) 1:100
- Mouse anti-human CD33-BB515 (Clone: WM53; BD Biosciences, 564588), Verified Reactivity: Human (QC Testing); Application: Flow cytometry (Routinely Tested). Dilution: 1:50
- Mouse anti-human CD7-BB700 (Clone: M-T701; BD Biosciences, 566488), Verified Reactivity: Human (QC Testing), Rhesus, Cynomolgus, Baboon (Reported); Application: Flow cytometry (Routinely Tested). Dilution: 1:50
- Mouse anti-human CD71-BV711 (Clone: M-A712; BD Biosciences, 563767), Verified Reactivity: Human (QC Testing); Application: Flow cytometry (Routinely Tested). Dilution: 1:20
- Mouse anti-human CD19-APCR700 (Clone: SJ25C1; BD Biosciences, 659121), Verified Reactivity: Human; Application: Flow cytometry. Dilution: 1:20
- Mouse anti-human CD3-BV605 (Clone: OKT3; Biolegend, 317322), Verified Reactivity: Human; Application: Flow cytometric analysis of antibody surface-stained cells. Dilution: 1:50
- Mouse anti-human CD56-PC5 (Clone: 5.1H11; Biolegend, 362516), Verified Reactivity: Human; Application: Flow cytometric analysis of antibody surface-stained cells. Dilution: 1:50
- Mouse anti-human CD34-BV421 (Clone: 561; Biolegend, 343610), Verified Reactivity: Human; Application: Flow cytometric analysis of antibody surface-stained cells. Dilution: 1:20
- Mouse anti-human CD1a-APC (Clone: HI149; BD Biosciences, 561755), Verified Reactivity: Human (QC Testing); Application: Flow cytometry (Routinely Tested) 1:33
- Mouse anti-human CD5-BUV737 (Clone: UCHT2; BD Biosciences, 612842), Verified Reactivity: Human (QC Testing); Application: Flow cytometry (Routinely Tested) 1:33
- Mouse anti-human CD42b-BV786 (Clone: HIP1; BD Biosciences, 740976), Verified Reactivity: Human (Tested in Development); Application: Flow cytometry (Qualified) 1:20
- Mouse anti-human CD41-PC7 (Clone: HIP8; Biolegend, 303718), Verified Reactivity: Human; Reported Reactivity: African Green, Baboon, Capuchin Monkey, Cynomolgus, Rhesus; Application: Flow cytometric analysis of antibody surface-stained cells. 1:50
- Mouse anti-human CD10-BV510 (Clone: HI10a; Biolegend, 312219), Verified Reactivity: Human, Cynomolgus, Rhesus; Reported Reactivity: African Green, Baboon, Capuchin monkey, Chimpanzee; Application: Flow cytometric analysis of antibody surface-stained cells. 1:100
- Mouse anti-human CD15-APCfire750 (Clone: W6D3; Biolegend, 323041), Verified Reactivity: Human; Application: Flow cytometric analysis of antibody surface-stained cells. 1:50

All the antibodies were purchased from Biolegend and BD Biosciences and they are well characterized and validated by providers. The stained samples were acquired through BD FACS Symphony A5 (BD Biosciences) cytofluorimeter after Rainbow bead (Spherotech) calibration.

Technically validated results were always included in the analyses, and we did not apply any exclusion criteria for outliers.

**Statistical analyses**
Statistical test and *p* values were specified in each figure legend or within figure graphs. To assess the correlation between two variables the Spearman index was calculated and a regression line (with 95% confidence intervals) was estimated after assessing the

normality of residuals as appropriate. Non-parametric tests were used for comparing two or more groups when continuous variables were considered, while the log-rank test was used for time-to-event data.

The evaluation of the longitudinal changes in the VCN measurements overtime was done by fitting mean response profiles models[65]. The proper variance-covariance matrix was selected within a set of competing forms (i.e., unstructured, autoregressive, heterogeneous autoregressive, compound symmetry) as the one that minimized the Akaike Information Criterion. When a significant interaction term (i.e. time x source) was detected, the comparisons of the two sources were performed at each time point. Data were eventually log-transformed in presence of a skewed distribution. All the tests were two-sided with $\alpha = 5\%$. The analyses were performed using Prism v9.1.0 software (GraphPad), except for the longitudinal models that were performed using SAS 6.4

### Reporting summary

Further information on research design is available in the Nature Portfolio Reporting Summary linked to this article.

## Data availability

Data supporting the current study are part of a registered clinical trial (NCT01515462). All the data generated in this study have been deposited in the San Raffaele Open Research Data Repository https://doi.org/10.17632/25v78pp9pg.1. These data are available under restricted access for the sensitive nature of the clinical data, access can be obtained by request to the corresponding author (Prof. Alessandro Aiuti, San Raffaele Telethon Institute for Gene Therapy (SR-TIGET), IRCCS San Raffaele Scientific Institute, Milan, 20132 Italy.). We intend to reply to any requests within two weeks and we will share the deposited data only for research purposes. The non-clinical data generated in this study are also provided in the Source Data file. Additional requests should be directed to the corresponding author. Source data are provided with this paper.

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

## Acknowledgements

This work was supported by Fondazione Telethon (TIGET Core Grant B2, A.A.), the Italian Ministero della Salute (Programma di Rete, NET-2011-02350069, A.A.), the European Commission (ERARE-3-JTC 2015 EURO-CID, A.A.) and Else Kröner-Fresenius-Stiftung (EKFS) prize (A.A.). We thank F. Ciceri, ME Bernardo and all medical and nursing staff of the Pediatric Immunohematology and Bone Marrow Transplantation Unit of the San Raffaele Scientific Institute and the San Raffaele Stem Cell Programme; S. Zancan, M. Casiraghi, G. Tomaselli and all Sr-TIGET Clinical Trial Office personnel for clinical trial management and support; the OTL team for revising the manuscript; M. Gabaldo and G. Farinelli for support with project management; and P. Massariello, G. Vallanti, M. Manfredini and other MolMed staff for patient cell manipulation. We thank C. Waskow for her advice in handling NSGW41 mice. We thank C. Villa, E. Canonico and S. Di Terlizzi of the Flow Cytometry Resource,

Advanced Cytometry Technical Applications Laboratory (FRACTAL) at Ospedale San Raffaele for cell sorting and technical help with instrumentation. We are indebted to the patients and their families for their commitment.

## Author contributions

S.S. performed phenotypic characterization, IS retrieval of HSPC progenitors and additional molecular testing for VCN estimation, performed in vitro and in vivo assays, collected and interpreted the data and wrote the manuscript. Fr.F. provided WAS patients' BM and PB samples and clinical data, L.B.-R. performed phenotypic characterization and isolation of HSPC subpopulations, performed in vitro and in vivo assays and analyzed the data. F.D. performed LAM–PCR and VCN estimation for all BM and PB patient samples. M.O. mapped IS and performed capture and re-capture analyses, P.Q. performed in vitro and in vivo assays for primitive HSPC populations. R.J.H. performed in vivo assay; L.D.C. designed and applied statistical analyses to normalize IS data and performed the exploratory analyses to assess the impact of each variables to IS dataset. F.B. performed IS retrieval through SLIM-PCR; I.M. and S.Gi. performed isolation of patient cell lineages. Fe.F., S.D. and E.A. contributed to clinical trial management and support. S.Ga. performed statistical analyses on longitudinal patients' samples. E.M. and A.C. supervised IS analyses and critically revised the manuscript. M.P.C. provided WAS patients' BM and PB samples and clinical data. A.A. contributed as PI by interpreting data, supervising the project and revising the manuscript.

## Competing interests

A.A. is the PI of the WAS-GT clinical trial sponsored by Orchard Therapeutics. F.F. and M.P.C. are investigators of the WAS-GT clinical trial sponsored by Orchard Therapeutics. All the other authors declare no competing interests.

## Additional information

Serena Scala[1], Francesca Ferrua[1,2], Luca Basso-Ricci[1], Francesca Dionisio[1], Maryam Omrani[1,3], Pamela Quaranta[1,4], Raisa Jofra Hernandez[1], Luca Del Core [1,5], Fabrizio Benedicenti[1], Ilaria Monti[1], Stefania Giannelli[1], Federico Fraschetta[2], Silvia Darin[2], Elena Albertazzi[2], Stefania Galimberti[6], Eugenio Montini [1], Andrea Calabria [1], Maria Pia Cicalese[1,2,4] & Alessandro Aiuti [1,2,4] ✉

[1]San Raffaele Telethon Institute for Gene Therapy (SR-TIGET), IRCCS San Raffaele Scientific Institute, Milan 20132, Italy. [2]Pediatric Immunohematology and Bone Marrow Transplantation Unit, IRCCS San Raffaele Scientific Institute, Milan 20132, Italy. [3]Department of Computer Science, Systems and Communication, University of Milano Bicocca, Milan 20126, Italy. [4]Università Vita-Salute San Raffaele, Milan 20132, Italy. [5]University of Groningen - Bernoulli Institute for Mathematics, Computer Science and Artificial Intelligence, Groningen 9747, Netherlands. [6]Center of Biostatistics for Clinical Epidemiology, University of Milano—Bicocca, Monza 20900, Italy. ✉e-mail: aiuti.alessandro@hsr.it

