## [Peer Review File · Nature Communications]

Hematopoietic reconstitution dynamics of mobilized- and bone marrow-derived human Hematopoietic stem cells after gene therapyReviewer #1 (Remarks to the Author):

This is a very well executed study by an excellent translational research team on the relative performance of two different sources of hematopoietic stem/progenitor cells (HSPC) in a gene therapy study for Wiskott-Aldrich syndrome. The data are clear, the analysis solid and the conclusions supported. The only issue in my mind is that the new information gained is limited and technical and therefore mostly of interest to transplant hematologists.

The main conclusion is that both sources perform equivalently for long term hematopoiesis but the rapidity of blood count recovery is somewhat better for mobilized peripheral blood (MPB) HSPC compared with bone marrow as the source of the graft. It has been well documented that MPB has that property. The new biology learned is not substantial and thus it will mostly guide the technical choice of HSPC source. The rigorously collected data are admirable, but unfortunately, the advance is rather incremental. It is not clear this will be of interest to the broad audience and broad biologic emphasis of Nat Communications.

Reviewer #2 (Remarks to the Author):

The authors describe a detailed comparison and analysis of hematopoietic reconstitution following lentiviral gene-modification of hematopoietic stem cells (HSPC) in patients treated with gene therapy (GT) for Wiskott-Aldrich Syndrome. In this study patients received conditioning chemotherapy followed by an infusion of purified, engineered CD34+ cells derived from either bone marrow (BM) or mobilized peripheral blood (MPB) where other variables/confounding factors remained constant (conditioning, vector, insert, transduction protocol, target cell dose, etc). The submitted manuscript describes a detailed characterization of transduction efficiencies, integration site analysis, engraftment kinetics, differentiation potential, longevity and lineage output of primitive and committed HSPC in 13 WAS-GT patients receiving either engineered MPB- or BM-HSPC.

The authors sought to dissect the behaviour and fate of gene-corrected BM and mobilized HSPC subpopulations, during early hematopoietic reconstitution after GT and after reaching the steady-state condition.

This is a unique study, which is of particular interest to the gene therapy community, but with wider implications for hematopoietic stem cell transplantation. It is almost certainly the largest longitudinal study comparing side by side GT treatment from two HSPC sources with an impressive level of detail including deep characterization of HSPC compartment after GT, molecular analyses of hematopoietic reconstitution and IS-based clonal tracking.

Main findings:

1. The authors demonstrated faster recovery from neutropenia and reduced time for platelet transfusion dependency in the MPB-GT group. This is consistent with published data in other settings and not particularly interesting. However, through advanced phenotypic characterisation of the CD34+ cell population they were able to analyse the impact of various primitive cell subsets within the CD34+ cell compartment at the time of transduction on subsequent reconstitution. This has not previously been examined. They identified that the number of myeloid progenitors and infused primitive HSPC, positively correlated with myeloid reconstitution and the transduced cell BM chimerism at steady state, respectively.
2. The distinct cell composition of the infused cell product was found to be the biggest determinant of early neutrophil reconstitution and long-term BM chimerism. This was supported by in vivo experiments using an established humanised mouse model, where the same number of adoptively transferred BM- or MPB- derived primitive HSPC populations resulted in comparable multi-lineage hematopoietic output and engraftment capability.
3. The main novel data in this study was generated through the ability to longitudinally track integration sites (IS) and clonal diversity in the infused gene-corrected cells. In this study the rapid engraftment observed in the MPB group was not associated with reduced clonality that has been demonstrated in other HSCT studies. Here, the cell product generated through the transduction of MPB retained increased cellular complexity and diversity compared to an 'equivalent' cell product generated from BM.
4. Finally, the authors demonstrated that a highly polyclonal pool of engineered HSPC could stably maintain the production of multi-lineage, gene-corrected cells for up to 3 years after GT in all the

patients analysed, despite the source of CD34+ cells, illustrating that both BM and MPB sources contain primitive and committed progenitors capable of efficiently supporting both short- and long-term hematopoiesis.

Specific Comments to the Authors:

Abstract: I am not clear what your conclusion is. Please clarify.

Introduction/Results: Please specify more clearly which patients were also included in the previous Scala et al Nature Medicine paper (2018) and what is the overlap – as some of these analyses seem to have been performed previously.

How do you explain the increased VCN observed in primitive cell subsets within the MPB group compared to the BM group?

Please explain in manuscript text what you mean by 're-captured' clone.

Figure 1.

1A; This lack of difference may be due to small sample size as trend to higher dose in MPB cf BM
Y axes 1D and 1E should say '% of'

Summary:

This paper is scientifically very sound. There are a few places where the English could be improved, but these are minor, and I have not specifically commented.

The data presented in the figures is generally described and interpreted accurately. The discussion elevates the paper and is well-referenced and thoughtful.

The data is novel in the sense that by using samples from patients treated on a GT trial (a unique and clinically relevant resource) they were able to carry out detailed phenotypic longitudinal analysis of multi-lineage hematopoietic reconstitution in patients who received MPB- or BM derived gene modified CD34+ cells. But some of these patients have previously been described in detail elsewhere (Scala S et al, Nat Med 2018) – including graft composition/transduction efficiencies and longitudinal analysis of engraftment (including IS studies).

Reviewer #3 (Remarks to the Author):

"Hematopoietic reconstitution dynamics of mobilized- and bone marrow-derived Hematopoietic stem cells after gene therapy" is an interesting work evaluating hematopoietic reconstitution kinetics, engraftment and clonality in 13 pediatric Wiskott-Aldrich syndrome patients treated with autologous lentiviral-vector transduced HSPC derived from MPB, BM or BM+MPB. The article, in my opinion, is well written and it explains in a exhaustive way the study.

The statistical methods used in the analysis are appropriate and well explained. One suggestion is to expand the statistical analysis in the text, describing also the methods used for detecting variable interactions (Shannon entropy, ...). Therefore a short summary of the methods presented in the supplementary materials should be also added in the main text.

Point by point reply

Reviewer #1 Comments:

This is a very well executed study by an excellent translational research team on the relative performance of two different sources of hematopoietic stem/progenitor cells (HSPC) in a gene therapy study for Wiskott-Aldrich syndrome. The data are clear, the analysis solid and the conclusions supported. The only issue in my mind is that the new information gained is limited and technical and therefore mostly of interest to transplant hematologists.

The main conclusion is that both sources perform equivalently for long term hematopoiesis but the rapidity of blood count recovery is somewhat better for mobilized peripheral blood (MPB) HSPC compared with bone marrow as the source of the graft. It has been well documented that MPB has that property. The new biology learned is not substantial and thus it will mostly guide the technical choice of HSPC source. The rigorously collected data are admirable, but unfortunately, the advance is rather incremental. It is not clear this will be of interest to the broad audience and broad biologic emphasis of Nat Communications.

We thank the reviewer for appreciating the quality of our work.

Concerning the criticisms about novelty arose by the reviewer we would like to point out specific novel aspects of our work:

1. This is the first work in which the performance of **purified CD34+ cells** from the two sources was evaluated. Indeed, the transplantation setting is largely based on infusion of unmanipulated or partially enriched total bone marrow or mobilized leukapheresis, and the effect of non-HSPC populations could have an important impact on the kinetic of engraftment. Moreover, no correlation between the composition of HSPC in the graft and the hematopoietic recovery was performed in the HSCT setting.
In our work instead we described, for the first time, that even purified MPB-derived CD34+ cells have different composition over the purified BM-CD34+ cells, and this difference correlates with the faster kinetic of myeloid reconstitution and transduced BM cell chimerism in MPB-GT patients. Finally, GT protocols rely on *ex vivo* manipulation of purified CD34+ cells before re-infusion into the patients and the direct translation of the information collected from the HSCT works was never proved.
2. In our work we compared the gene therapy **outcome of BM vs. MPB G-CSF+Plerixafor CD34+ cells**. Previous longitudinal HSCT studies were mainly focused on comparing the outcome of unmanipulated whole BM vs. unmanipulated whole MPB (mobilized by G-CSF only) in terms of overall survival, incidence of graft versus host disease and relapse or immune reconstitution (Körbling and Anderlini, 2001; Waller et al., 2019). No information exists on the hematopoietic reconstitution of MPB product after G-CSF + Plerixafor mobilization in comparison with BM-based HSCT.
3. In our study we performed a **comparison in the absence of major confounding factors**. The vast majority of BM vs. G-MPB HSCT evaluations were often performed in heterogeneous groups of patients with distinct type of disease background/hematological malignancies, conditioning, prior treatments and transplantation setting (ie autologous vs. heterologous). In our manuscript instead we had the unique opportunity to analyze the performance of the two sources in a homogenous group of patients in terms of range of age, disease background, transduction protocol and conditioning regimen. Moreover, in our setting the

hematopoietic reconstitution was not biased by administration of additional treatments after infusion of gene-corrected cells. Indeed, in the clinical practice, patients undergoing HSCT often are administered post-infusion with growth factors (G-CSF) to speed up myeloid recovery, introducing an additional confounding factor in the kinetic of hematopoietic reconstitution. Thus, the uniqueness of our cohort of patients allowed us to perform the cleanest comparison of the two sources in the context of gene therapy and transplantation.

4. The reviewer is right that one of the findings of our manuscript is that MPB CD34+ cells showed faster recovery with respect to BM CD34+ cells and that the two sources performed equally in sustaining the long-term hematopoiesis. As mentioned above, this finding was not obvious and it implies that the *in vitro* manipulation per se does not introduce changes in the overall characteristics of CD34+ cells. Nevertheless, this is not the only conclusion of our work. As mentioned in the discussion (lines 285-295) one of the most important and original aspects of our analyses is **the correlation of the CD34+ cell composition** undergoing *in vitro* manipulation **with the outcome of the gene therapy procedure**. Our study shows that the two most relevant parameters were the amount of myeloid progenitors and the number of the infused primitive HSPC, positively correlating with the myeloid reconstitution and the transduced cell BM chimerism at steady state, respectively. Finally, the possibility of **tracing the dynamics of infused gene-corrected HSPC through integration site** (lines 316-324) provided substantial novel information over previous HSCT studies, where cells contained in the graft are not marked. Indeed, to our knowledge, no study has been conducted to evaluate whether the rapid hematopoietic recovery observed in MPB-based HSCT might have occurred at the expenses of the overall clonality of the hematopoietic system. In our unique model, we could assess that the use of MPB as source was associated with an overall higher complexity of the hematopoietic system and diversity of HSPC pool with respect to BM CD34+ cells. This novel observation is particularly relevant in the clinical practice since it was reported that reduction of HSPC clonality may lead to premature cellular senescence in HSCT.

Overall, the conclusions of our work provided substantial novel information over the previous literature with broad interest in basic, translational and clinical research.

First of all, our work has important implication in **hematopoietic stem cell biology**, indicating that differential transcriptional signature observed in different stem cell sources (including cell cycle activity, stemness associated modules) might be explained by differential CD34+ cell composition, rather than intrinsic differences. Additionally, since we cannot completely exclude that transcriptionally differences exist between the two sources, our data suggest that these distinct profiles might not result in differential functional properties of CD34+ cell populations. Indeed, the equal performance of primitive HSPC subsets *in vivo* (in both GT -treated patients and xenotransplanted mice) suggest that the transcriptionally different populations might respond similarly in highly stressed environment, such as the one present upon transplantation.

Second, our results have important translational readouts. In the **pre-clinical development**, cord blood is often used as source of stem cells to evaluate the performance of *ex vivo* or *in vivo* treatments. However, given the impact of CD34+ cell composition in the engraftment kinetic, chimerism in the long-term and clonality, the translatability of these information to other HSPC sources should be carefully assessed. Moreover, our study suggests that a more advanced phenotypic characterization of the HSPC before *ex vivo* manipulation may be predictive of the *in vivo* outcome.

Finally, as also stated by reviewer #2, our work is “of particular interest to the **gene therapy community**, but with wider implications for hematopoietic stem cell transplantation”. Since GT approaches are widely exploited not only to treat hematological disorders but also metabolic diseases, including Metachromatic Leukodystrophy and Mucopolysaccharidosis, as well as solid tumors our study has relevant **clinical implications**, beyond the field of hematology. Understanding the impact of the CD34+ cell composition on the kinetic of reconstitution, the transduced cell chimerism and the clonality of the graft will not simply “guide the technical choice of HSPC source” but would rather improve and widen the application of HSPC-based therapies.

We have now included some of these novelty points in the manuscript (Introduction and Discussion sections), in order to stress the importance of our work.

Reviewer #2 Comments:

The authors describe a detailed comparison and analysis of hematopoietic reconstitution following lentiviral gene-modification of hematopoietic stem cells (HPSC) in patients treated with gene therapy (GT) for Wiskott-Aldrich Syndrome. In this study patients received conditioning chemotherapy followed by an infusion of purified, engineered CD34+ cells derived from either bone marrow (BM) or mobilized peripheral blood (MPB) where other variables/confounding factors remained constant (conditioning, vector, insert, transduction protocol, target cell dose, etc).

The submitted manuscript describes a detailed characterization of transduction efficiencies, integration site analysis, engraftment kinetics, differentiation potential, longevity and lineage output of primitive and committed HSPC in 13 WAS-GT patients receiving either engineered MPB- or BM-HSPC.

The authors sought to dissect the behaviour and fate of gene-corrected BM and mobilized HSPC subpopulations, during early hematopoietic reconstitution after GT and after reaching the steady-state condition.

This is a unique study, which is of particular interest to the gene therapy community, but with wider implications for hematopoietic stem cell transplantation. It is almost certainly the largest longitudinal study comparing side by side GT treatment from two HSPC sources with an impressive level of detail including deep characterization of HSPC compartment after GT, molecular analyses of hematopoietic reconstitution and IS-based clonal tracking.

Main findings:

1. The authors demonstrated faster recovery from neutropenia and reduced time for platelet transfusion dependency in the MPB-GT group. This is consistent with published data in other settings and not particularly interesting. However, through advanced phenotypic characterisation of the CD34+ cell population they were able to analyse the impact of various primitive cell subsets within the CD34+ cell compartment at the time of transduction on subsequent reconstitution. This has not previously been examined. They identified that the number of myeloid progenitors and infused primitive HSPC, positively correlated with myeloid reconstitution and the transduced cell BM chimerism at steady state, respectively.
2. The distinct cell composition of the infused cell product was found to be the biggest determinant of early neutrophil reconstitution and long-term BM chimerism. This was supported by in vivo experiments using an established humanised mouse model, where the same number of adoptively transferred BM- or MPB- derived primitive HSPC populations resulted in comparable multi-lineage hematopoietic output and engraftment capability.
3. The main novel data in this study was generated through the ability to longitudinally track integration sites (IS) and clonal diversity in the infused gene-corrected cells. In this study the rapid

engraftment observed in the MPB group was not associated with reduced clonality that has been demonstrated in other HSCT studies. Here, the cell product generated through the transduction of MPB retained increased cellular complexity and diversity compared to an 'equivalent' cell product generated from BM.

4. Finally, the authors demonstrated that a highly polyclonal pool of engineered HSPC could stably maintain the production of multi-lineage, gene-corrected cells for up to 3 years after GT in all the patients analysed, despite the source of CD34+ cells, illustrating that both BM and MPB sources contain primitive and committed progenitors capable of efficiently supporting both short- and long-term hematopoiesis.

We thank the reviewer for his/her comments and for acknowledging the scientific advances of our work. Please find below the reply to the specific comments.

Specific Comments to the Authors:

Abstract: I am not clear what your conclusion is. Please clarify.

We thank the reviewer for pointing out this aspect. We have now updated the abstract in order to be more focused on the specific conclusions of the work.

Introduction/Results: Please specify more clearly which patients were also included in the previous Scala et al Nature Medicine paper (2018) and what is the overlap – as some of these analyses seem to have been performed previously.

We would like to specify that only 6 out of 14 patients (Pt1 to Pt6) included in this study were present in the work published on Nat Med in 2018. Although, as correctly pointed out from the reviewer, some of the data included in the current manuscript were also present in the Nat Med work (such as the CD34+ composition before GT, the longitudinal count of mature myeloid and lymphoid populations and VCN of HSPC subsets), the purpose of the work was different. Specifically, in the Nat Med paper we analyzed 6 patients (1 BM+MPB-GT and 5 BM-GT) to investigate the role of the distinct HSPC subpopulations in supporting the distinct phases of hematopoietic reconstitution after GT. In the present Nat Comm manuscript instead, we expanded the dataset of the first 6 patients, adding new data, and enlarged our cohort with 8 additional patients treated with MPB CD34+ cells, with similar age with respect to the published BM-GT group.

In particular, in addition to generate the entire dataset for MPB-GT patients, we included the early myeloid and platelet reconstitution (Fig2), longitudinal VCN in mature subpopulations (Fig 4A-G), percentage of transduction and mean VCN of BM colonies isolated both at early and late phases (Fig 4 L-M). In addition, no correlation with the graft composition was performed in the Nat Med paper, while in this manuscript we detailed for the first time, the impact of the CD34+ cell composition on the gene therapy outcome. Moreover, we performed a specific analysis to assess the number of lentiviral vector integrated copies to estimate the permissiveness of the two sources to the transduction procedures (Supplementary Fig E-H). Finally, to estimate more precisely the diversity and abundance of engrafted clones, we re-generated the integration site dataset of both BM-GT, BM+MPB-GT and MPB-GT patients' subsets through SLIM-PCR (Fig 5A-E and Supplementary Fig.8A-C). Thus, we used the previous Nat Med IS dataset only for the assessment of the lineage commitment of primitive HSPC subsets in both groups of patients (Fig 5G and Supplementary Fig 8D).

We already reported some of these information in the Supplementary Material section, but we have now expanded this aspect in the revised manuscript in both the Main and the Supplementary Methods sections to better clarify this point.

How do you explain the increased VCN observed in primitive cell subsets within the MPB group compared to the BM group?

We thank the reviewer for pointing out the lack of clarity for this data.

Since WAS-GT patients underwent sub-myeloablative conditioning before infusion of gene-corrected cells, the hematopoietic compartment within the BM is composed by a mixture of transduced and non-transduced cells. Having observed that the two groups of patients had similar mean VCN of BM colonies and comparable distribution in the number of integrated copies within the single colonies we ruled out that the higher VCN in primitive HSPC subsets in MPB-GT was the result of higher number of integrated copies. This finding instead is due to the higher BM cell chimerism observed in the MPB-GT group. Indeed, in case of the same number of integrated copies (as observed in BM-GT and MPB-GT groups), an increased amount of gene-corrected cells (found in MPB-GT patients) would result in increased mean VCN for the analyzed population.

We have now better clarified this point in the main text, also modifying the order of the panels in Figure 4.

Please explain in manuscript text what you mean by 're-captured' clone.

We had originally thought to include the explanation of the re-capture approach in the text, but we moved it to the supplementary method section to fit within the word counts required for Nar Comm. We have now replaced the explanation in the main text.

Figure 1.

1A; This lack of difference may be due to small sample size as trend to higher dose in MPB cf BM
Y axes 1D and 1E should say '% of'

As the reviewer correctly pointed out there is a trend over increased CD34+ cell dose in MPB-GT patients. This is mainly due to the higher number of CD34+ cells that can be retrieved from MPB vs BM collection, resulting in increased number of CD34+ cells transduced and infused.

However, the purpose of the Fig 1A was to show that in our analyzed cohort of patients, this trend was not statistically significant, while the amount of infused primitive and myeloid subpopulations was found to have high statistical significance. Moreover, we also observed that CD34+ cell dose did not correlate with both early myeloid reconstitution (Supplementary Fig 4) as well as with the transduced BM chimerism in the long term (Fig 4J), implying that the total CD34+ cell dose was not able to fully explain the differential performance of the two sources *in vivo* upon infusion.

We have now included in the revised manuscript the observation that "despite we observed a trend to higher CD34+ cell dose, this was not statistically significant." Moreover, we modified the axis labels according to the reviewer suggestion.

Summary:

This paper is scientifically very sound. There are a few places where the English could be improved, but these are minor, and I have not specifically commented.

The data presented in the figures is generally described and interpreted accurately. The discussion elevates the paper and is well-referenced and thoughtful.

The data is novel in the sense that by using samples from patients treated on a GT trial (a unique and clinically relevant resource) they were able to carry out detailed phenotypic longitudinal analysis of multi-lineage hematopoietic reconstitution in patients who received MPB- or BM derived gene modified CD34+ cells. But some of these patients have previously been described in detail elsewhere (Scala S et al, Nat Med 2018) – including graft composition/transduction efficiencies and longitudinal analysis of engraftment (including IS studies).

We thank the reviewer for his/her positive comments on the importance of our work. Concerning the overlap of the dataset between this manuscript and the published Nat Med paper, please refer to the comment above and to the revised manuscript.

Reviewer #3 (Remarks to the Author):

“Hematopoietic reconstitution dynamics of mobilized- and bone marrow-derived Hematopoietic stem cells after gene therapy” is an interesting work evaluating hematopoietic reconstitution kinetics, engraftment and clonality in 13 pediatric Wiskott-Aldrich syndrome patients treated with autologous lentiviral-vector transduced HSPC derived from MPB, BM or BM+MPB. The article, in my opinion, is well written and it explains in a exhaustive way the study.

The statistical methods used in the analysis are appropriate and well explained. One suggestion is to expand the statistical analysis in the text, describing also the methods used for detecting variable interactions (Shannon entropy, ...). Therefore a short summary of the methods presented in the supplementary materials should be also added in the main text.

We are grateful to the reviewer for his/her positive response. We have included some of the supplementary methods within both the main text and the main Materials and Methods section.

Reviewer #1 (Remarks to the Author):

The authors have better clarified the novel aspects of their work. While it remains mostly incremental the quality and depth of analysis is of use to the gene therapy field.

Reviewer #2 (Remarks to the Author):

The authors have addressed my queries appropriately.

I retain concerns regarding the broader relevance of this manuscript and the degree to which it advances the field.

Reviewer #3 (Remarks to the Author):

The Authors have replied to my questions and followed my suggestion.